# Investigation into instantaneous centre of rotation for enhanced design of floating offshore wind turbines

Katarzyna Patryniak[1], Maurizio Collu[1], Jason Jonkman[2], Matthew Hall[2], Garrett Barter[2], Daniel Zalkind[2], and Andrea Coraddu[3]

[1]Department of Naval Architecture, Ocean and Marine Engineering, University of Strathclyde, Glasgow, UK
[2]National Renewable Energy Laboratory, Golden, CO, USA
[3]Faculty of Mechanical, Maritime and Materials Engineering, Delft University of Technology, Delft, Netherlands
Correspondence: Katarzyna Patryniak (katarzyna.patryniak@strath.ac.uk)

**Abstract.** The dynamic behaviour of floating offshore wind turbines (FOWTs) involves complex interactions of multivariate loads from wind, waves, and currents, which result in complex motion characteristics. Although methods for analysing global motion responses are well-established, the time- and location-dependent kinematics remain underexplored. This paper investigates the instantaneous centre of rotation (ICR), a point of zero velocity at a time instance of general plane motion. Understanding and strategically positioning the ICR can reduce the dynamic motion in critical structural locations, enhancing the performance and structural robustness of FOWTs. The paper presents a method for computing the ICR using time-domain simulation results and proposes a statistical analysis approach suitable for design studies. Building on prior research, it examines the sensitivity of the ICR to external loading and design features, providing insights into how these factors influence motion response and how the motion response influences the statistics of the ICR, structural loads, and other performance metrics of interest. The study explores two FOWT configurations, a spar and a semisubmersible, identifying design variables that most effectively control the ICR statistics and identifying the ICR statistics most correlated with the responses of interest. Finally, through two case studies, we demonstrate how to apply these new insights in a practical design scenario. By adjusting the design variables most correlated with the ICR (fairlead vertical position and centre of mass for the spar, and mooring line length and offset column diameter for the semisubmersible), we successfully modified the designs of the floating support structures to reduce the loads in the mooring lines, tower base, and blade roots, improving the ultimate strength and fatigue characteristics as compared to the original designs.

## 1 Introduction

The dynamic behaviour of floating offshore wind turbines (FOWTs) presents significant complexity due to the diverse range of multivariate loads they encounter in an offshore environment. The distinctly distributed inertial, hydrostatic, mooring, and aero- and hydrodynamic loads are influenced by varying combined wind, wave, and current impacts. This results in response characteristics unique to FOWTs and not observed in other offshore structures. Although the methods for analysing the global motion responses of FOWTs are well established, the intricate, time- and location-dependent kinematics and motion patterns remain underexplored.

As highlighted by Patryniak et al. (2023), when a FOWT undergoes general plane motion, one can identify a point, known as the instantaneous centre of rotation (ICR), where the velocity is zero at a given moment. This point is not necessarily located within the physical boundaries of the FOWT and is highly dynamic, shifting in response to changing environmental loads and the resulting platform motions. Accurate identification of the ICR and its statistics can offer valuable insights for optimising motion reduction strategies and enhancing the FOWT's performance and structural robustness. By strategically modifying design elements to influence the position of the ICR, it is possible to minimise dynamic motions at critical locations. For example, aligning the fairleads with the ICR could limit the dynamic part of the motion that does not contribute to useful restoring but does contribute to fatigue damage of the mooring lines. Similarly, positioning the ICR near the rotor-nacelle assembly (RNA) could decrease rotor and nacelle motion, reducing aerodynamic load fluctuations and mitigating fatigue across the system.

Early studies on the centre of rotation of floating structures were conducted by Stewart and Ewers (1979) and Standing (1991), focusing on barge vessels. Both authors noted significant oscillations in the centre of rotation, with Standing observing that the point "tends to wander wildly" in irregular waves. This work challenged the assumption that the centre of rotation is aligned with the centre of gravity, and highlighted the importance of hydrodynamic loads. Haslum and Faltinsen (2000) further examined the centre of rotation in a spar platform undergoing surge-pitch motion, revealing its sensitivity to oscillation frequency and fairlead location. Souza et al. (2012) introduced an experimental method to determine the ICR using velocity vectors from two probes on the floating body. Fernandes et al. (2016) and Costa et al. (2018) built on the previous work by analysing a floating production storage and offloading (FPSO) vessel in regular waves, showing that the ICR locus (a curve formed by all ICR points) is a straight line that does not align with the vessel's symmetry and varies with wave frequency. At very low frequencies, the ICR exhibited a behaviour akin to pure translation, approaching an infinite distance above the body. Costa et al. (2020a) found that the ICR coordinates of a moored vessel in regular head waves, free to move in surge, heave, and pitch, closely follow a Cauchy distribution. In subsequent work (Costa et al., 2020b), the authors expanded their investigation to oblique seas and six degrees of freedom, further reaffirming the ICR's dependence on external loading frequency.

Despite its relevance, the ICR has not been a focal point in FOWT studies. A few examples include the work by Eliassen (2015), which was the first attempt to investigate the centre of rotation of a FOWT without presupposing its location, and the study by Kaptan et al. (2022), which examined the influence of wave frequency on the centre of rotation for two different floating concepts. Lemmer et al. (2020) considered the centre of rotation as the point where the surge and pitch response amplitude operators combine to produce near-zero motion. The authors modified the semisubmersible floating platform designs to ensure that the system rotates about the hub, thereby achieving preferable dynamic response characteristics, improved power quality, and reduced tower base loads. While the location of the centre of rotation was determined through frequency domain analysis, its instantaneous nature was not taken into account.

The concept of ICR in the context of FOWTs was most recently explored by Patryniak et al. (2023). The study provided a comprehensive explanation of the ICR concept, developed a method for calculating the ICR of FOWTs through eigenanalysis and in the time domain, and analysed its time history and statistical behaviour for a specific case involving the OC3 Hywind spar (Jonkman, 2010) with the NREL $5\,\mathrm{MW}$ reference rotor (Jonkman et al., 2009), subjected to regular and irregular waves

of varying frequencies and amplitudes[1]. The findings indicated that the ICR of a FOWT is influenced by the frequency of the incident waves and, at low frequencies, is also affected by wave height. Although significant insight into the ICR behaviour was provided, only one floating structure was investigated, and realistic environmental conditions (combined wind, wave, and current load) were not considered, limiting the applicability of the study.

In this work, we build on the previously developed method and further explore the ICR for FOWT design variations under complex loading conditions typical of operational FOWTs to incorporate this knowledge into the design process. We begin by examining the influence of more complex environmental loading conditions (Sect. 3) to enhance understanding of ICR behaviour in realistic operational scenarios. For ICR insights to be useful in the design process, two conditions must be met. First, the ICR must be correlated with FOWT responses that are critical for design improvement. Second, a set of design variables that can be adjusted to "control" the ICR statistics must be identified to enable these improvements. Accordingly, in Sect. 4, we explore how specific design decisions affect ICR behaviour, followed by an investigation of the relationship between the ICR and key FOWT responses (Sect. 5). This sensitivity analysis helps identify key design features that can be prioritised during the FOWT optimisation process. The methods proposed by this paper can be applied in a practical design scenario, as will be demonstrated in Sect. 6.

## 2    Methodology

The objectives of this study are i) to understand the behaviour of the ICR in a complex, realistic environment, and ii) to demonstrate how knowledge of ICR behaviour can be leveraged to improve FOWT designs. The ICR can serve as a useful metric in the FOWT design process, provided that:

1. The FOWT design can be modified in ways that influence the ICR statistics.

2. The ICR statistics significantly affect the responses of interest (i.e., the performance of the FOWT).

Therefore, to enable a practical application of the new knowledge about the ICR of FOWTs, this study examines the relationships among design variables, ICR statistics, and response statistics. In particular, we investigate which design variables are significantly correlated with the ICR statistics and which ICR statistics are significantly correlated with the responses of interest. The procedure followed is illustrated in Fig. 1 and detailed in the subsequent sections.

The study begins with one-at-a-time perturbations of multiple design variables, which are features of the floating platform and mooring system. This approach allows for the examination of correlations with ICR statistics (mean and standard deviation of horizontal and vertical ICR components). Although more comprehensive experimental designs, such as full factorial or Latin hypercube, offer deeper insights into design space behaviour, perturbing one variable at a time while keeping all other variables at their original design values significantly reduces the design space size, keeps the computational cost tractable, and allows for clear presentation and interpretation of the results.

---

[1]OC3: Offshore Code Comparison Collaboration

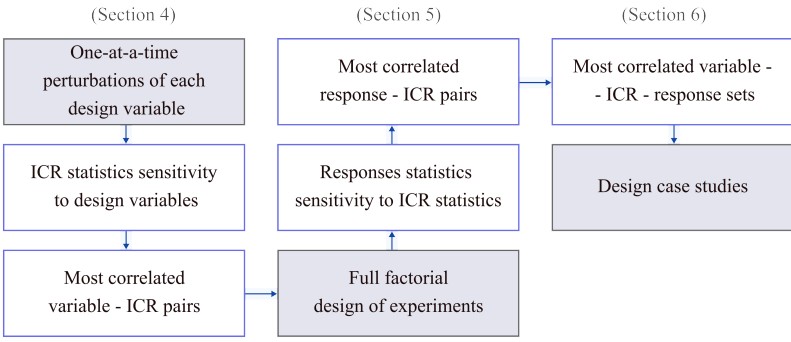

**Figure 1.** The procedure for investigating the relationships between the design variables, ICR, and response statistics.

Once the relationships between the design variables and ICR are established, the most impactful parameters are identified for downselection. These parameters are then subjected to a more thorough sensitivity analysis using a full factorial approach. This analysis explores the relationships between all combinations of the selected parameters and the responses of interest (i.e., design performance metrics).

At this stage, highly correlated pairs of design variables and ICR statistics are further refined to include only those that also show strong correlations with the responses of interest. This systematic procedure helps identify design directions that may lead to significant improvements in the performance of the FOWTs through changes to ICR statistics.

The methods for computing the ICR and its statistics are outlined below.

## 2.1 Instantaneous centre of rotation derivation

When exposed to colinear unidirectional wind, wave, and current loads, FOWTs undergo general plane motion, with translational and rotational characteristics dependent on the external load and design of the structure (Patryniak et al., 2023). In particular, the centre of the pitch rotation depends on the amplitudes, frequencies, phases, and distributions of the external dynamic forces, and therefore its determination is nontrivial.

A point (not necessarily within the body) of zero velocity at a particular instant in time (i.e., ICR) can be found by analysing the velocity vectors at two arbitrarily chosen points on the body. While, by definition, the ICR appears to be instantaneously at rest, all other points in the body present pure rotation about this point, i.e., they follow circular paths around the ICR (Meriam and Kraige, 1993). Therefore, the ICR can be found at the intersection of the perpendiculars to the velocities of the two points ($A$ and $B$ in Fig. 2), assuming a rigid body. In the special case when the velocity vectors are parallel and do not cross, and the line joining points $A$ and $B$ is perpendicular to the velocities, the ICR can be obtained through a direct proportion[2] (Meriam and Kraige, 1993), as illustrated in the middle subfigure of Fig. 2. In the special case of pure translation (right subfigure of Fig. 2), the ICR tends to infinity. The vectors required for this calculation can be found by measuring the velocities in $A$ and $B$, as obtained from the time-domain numerical simulations (detailed in Sect. 2.3).

---

[2]Direct proportion is a relation between two quantities where the ratio of the two is equal to a constant value.

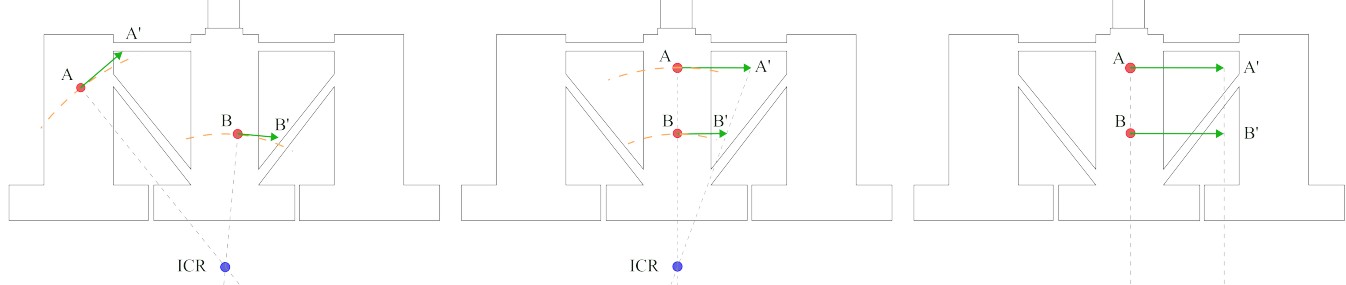

**Figure 2.** Construction of the ICR using velocity vectors at two points on the floating body. Left: general case, misaligned nodes, middle: special case, aligned nodes, right: special case, pure translation.

Note that the location of the ICR changes as the body moves or loading conditions change, both in terms of the body-fixed and global inertial coordinate systems, and therefore, it only exists at a particular instant in time (hence, "instantaneous"). As shown previously by Patryniak et al. (2023) and reiterated here in Fig. 3, in regular waves (a purely theoretical condition), the ICR coordinates exhibit a distinctive time history pattern. When the pitch displacement reaches its minimum or maximum, the pitch velocity is zero (pure translation), and the ICR tends toward infinity. At other times, it concentrates around finite locations. The phases before and after each pitch extremum result in the ICR clustering around unique points, as also illustrated in the histograms of Fig. 4. In a more complex, stochastic environment, such a regular pattern cannot be observed (Fig. 5). However, still, ICR coordinates follow a well-defined distribution illustrated in Fig. 6.

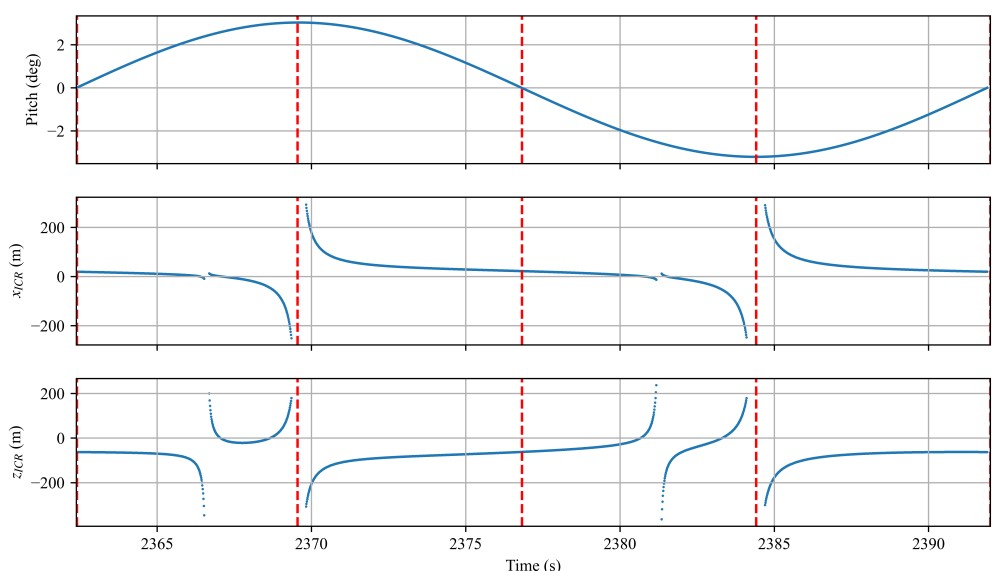

**Figure 3.** ICR time history during one period of pitch motion. Reproduced from Patryniak et al. (2023).

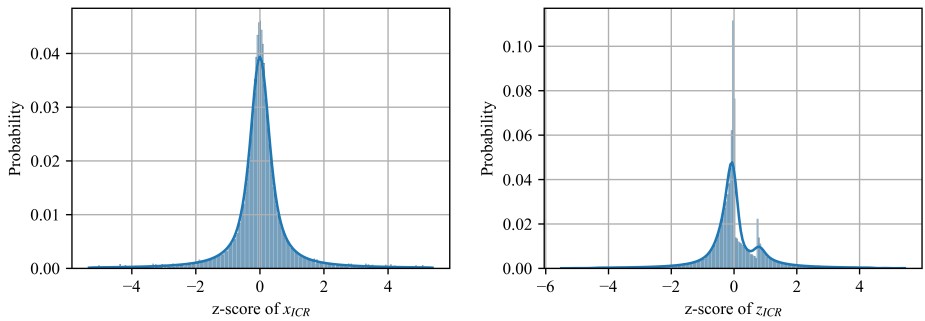

**Figure 4.** Histogram of ICR coordinates for a FOWT in regular waves.

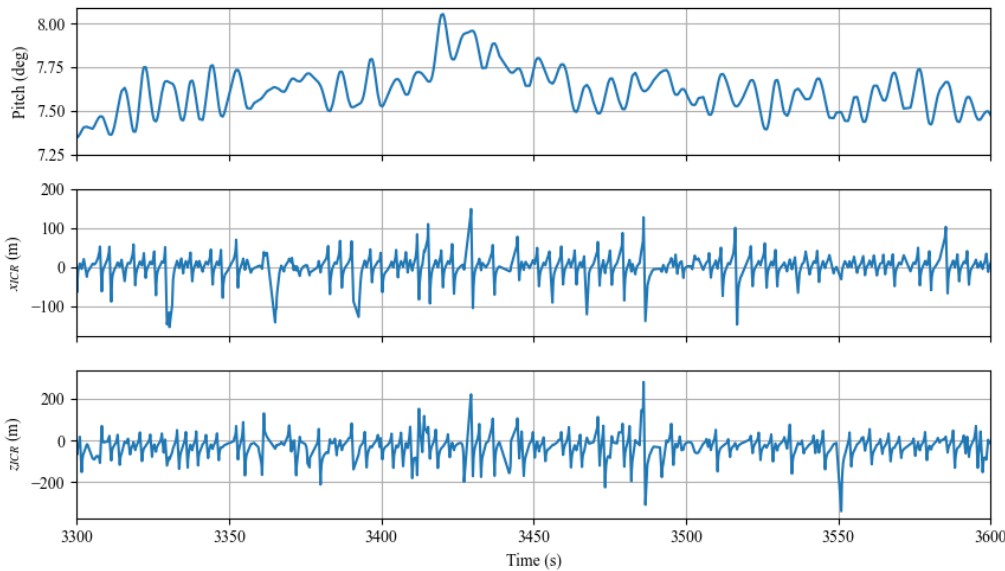

**Figure 5.** ICR time history during platform motion in a realistic environmental condition.

## 2.2 Instantaneous centre of rotation statistics

Given that the ICR can exhibit asymptotic behaviour and is very irregular under stochastic environmental loading conditions, analysing time-domain signals directly can be difficult and not readily applicable in the design context. Therefore, a statistical analysis approach is followed instead. In particular, the distribution characteristics of the in-plane coordinates of the ICR (type, mean, standard deviation, skewness and kurtosis) are investigated.

One-sample Kolmogorov-Smirnov (K-S) test with hypothesis testing (Hodges, 1958) is performed to establish whether the

125 ICR follows a normal distribution. This test compares data sampled from a fitted normal distribution with the samples from the original data and outputs two metrics: the K-S statistic, which measures the maximum discrepancy between the empirical cumulative distribution functions of the two samples, and the p-value, which indicates the probability of observing the test

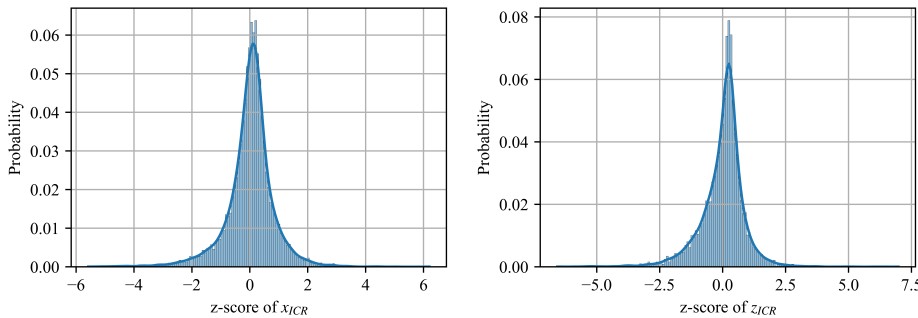

**Figure 6.** Histogram of ICR coordinates for a FOWT in a realistic environmental condition.

results under the null hypothesis. The null hypothesis is that the two samples come from the same distribution. A high p-value above the significance level of $0.05$ suggests insufficient evidence to state that the two samples come from different distributions. Therefore, a low K-S value and a p-value above $0.05$ are needed to confirm that the data closely follow a normal distribution.

To analyse the relationships between design variables, the ICR, and FOWT responses, we compute the Spearman rank correlation coefficient[3], which quantifies the statistical dependence between two variables. As illustrated in Table A1 in Appendix A, for each pair of variables (for example, the standard deviation of the tower base bending moment and the standard deviation of $z_{ICR}$), we list their values from different simulations, then rank each list from smallest to largest (the smallest value in a list gets rank 1, the second smallest value gets rank 2, etc.). The Spearman correlation coefficient $\rho$ is computed based on the sum of the differences between the ranks of the paired values:

$$\rho = 1 - \frac{6 \sum_{i=1}^{n} d_i^2}{n(n^2 - 1)} \tag{1}$$

$d_i$ is the difference between the ranks for sample (simulation) $i$, and $n$ is the number of observations (simulations). Spearman's correlation assesses the strength of a monotonic relationship, making no assumptions about the underlying data distribution. A correlation close to $+1$ or $-1$ indicates a strong monotonic relationship:

- $\rho \approx 1$: as one variable increases, the other tends to increase.

- $\rho \approx -1$: as one variable increases, the other tends to decrease.

- $\rho \approx 0$: little to no monotonic relationship between the variables.

Since Spearman's correlation is based on ranks rather than raw values, it is robust to nonlinearity and less sensitive to outliers than Pearson's correlation. However, it does not imply causation, meaning that a strong correlation does not indicate that changes in one variable directly cause changes in another.

---

[3]The rank of a value is its position within a sorted list of data points, ordered from smallest to largest.

## 2.3 Dynamic simulations

All simulations in this study are performed with OpenFAST, a state-of-the-art, extensively validated nonlinear aero-hydro-servo-elastic coupled time-domain model of dynamics (NREL, 2024a). Equations of motion of OpenFAST are nonlinear and expressed in a modular generalised state-space form with module-level states and module-to-module coupling handled through input-output constraints (Jonkman, 2013).

The degrees of freedom include six rigid body motion modes; although possible in OpenFAST, the elastic dynamics are not modelled in this study, as the analysis relies on a rigid body assumption. The NREL 5 MW baseline variable-pitch variable-speed controller allows for blade pitch and rotor speed variations depending on the mean wind speed and slight rotor speed variations due to the system motion and turbulent inflow. The aerodynamic solution relies on the blade-element momentum theory with Beddoes-Leishman unsteady blade airfoil aerodynamics model, tip and hub loss corrections, and tower influence based on the potential flow. The first-order potential hydrodynamics of the spar platform are modelled through frequency-to-time-domain transforms based on the potential coefficients obtained from the boundary element method (BEM) code WAMIT (Lee and Newman, 2006). Viscous loads are computed from Morison's theory. The semisubmersible platform's offset columns, heave plates, and main column are modelled using a hybrid approach (potential and Morison). The potential coefficients are obtained with pyHAMS BEM solver (NREL, 2024b), as implemented in RAFT (Hall et al., 2022). The slender pontoons and cross-braces (diameter of 1.6 m) are treated with a Morison-only approach, with hydrodynamic coefficients listed in Table 1. Second-order hydrodynamics are not considered.

**Table 1.** Hydrodynamic model parameters.

| Parameter | Unit | OC3 spar | OC4 semisubmersible |
|---|---|---|---|
| Maximum Morison element size | m | 0.5 | 1 |
| Maximum BEM panel size | m | 1 | 1.5 |
| Added mass coefficient | - | - | Pontoons and cross-braces: 0.63 |
| Transverse viscous-drag coefficient | - | 0.6 | Main column: 0.56 |
| | | | Offset column: 0.61 |
| | | | Heave plate: 0.68 |
| | | | Pontoons and cross-braces: 0.63 |

The nonlinear mooring loads are computed using a lumped-mass mooring line model, MoorDyn (Hall and Goupee, 2015). The model includes the effects of axial stiffness and damping, weight and buoyancy, hydrodynamic viscous forces, and vertical spring-damper forces from contact with the seabed.

In this work, the design variations involving a change of the underwater shape of the floating platform are kept small to avoid the need for repeated solution of the potential coefficients, which is computationally expensive. For large-volume structural members, this may still impact the accuracy of the solution. Therefore, the results in this paper are interpreted assuming no change in potential hydrodynamic loads.

Section 3 explores the ICR under various synthetic environmental conditions. Instead of relying on the predefined design load cases (e.g., DLC 1.6, which involves an operational turbine in severe sea state, and DLC 6.1, which considers a parked turbine experiencing severe wind and sea conditions, as outlined in the IEC 61400-3-2 standard (IEC, 2019)), this study selectively applies different loading components. This approach aims to provide deeper insights into the ICR behaviour related to specific sources of loading. The wind conditions are characterised by the mean hub-height wind speed, profile (shear), and turbulence intensity, and the wave conditions (sea state) are characterised by the (significant) wave height and (peak) period. Wind and wave directions are aligned with the positive $x$-axis (no misalignment is considered in this paper). The wave peak-shape parameter is set based on the peak period and significant wave height, as recommended in the IEC 61400-3 Annex B (IEC, 2009).

Sections 4–6, on the other hand, investigate ICR over a range of realistic environmental conditions. Each design variant is simulated in 10 environmental conditions, listed in Table 2, representing binned metocean data obtained for the Scottish sectoral marine plan for the NE8 site (Scottish Government, 2020) (these bins account for $98.4\,\%$ of cumulative probability). The responses presented in this paper refer to the average responses weighed with the probability of each of the 10 conditions. Each simulation is run for $1\,\mathrm{hour}$ (excluding the initial transient phase) with a single random seed (the same seed in all simulations).

**Table 2.** Environmental conditions binning. Wind shear fixed at 0.2 and turbulence intensity at $14\,\%$.

| Condition | 1 | 2 | 3 | 4 | 5 | 6 | 7 | 8 | 9 | 10 |
|---|---|---|---|---|---|---|---|---|---|---|
| $V_s$ ($\mathrm{m\,s^{-1}}$) | 4.1 | 6.3 | 8.5 | 10.7 | 12.9 | 15.1 | 17.3 | 19.5 | 21.7 | 23.9 |
| $H_s$ (m) | 1.09 | 1.23 | 1.44 | 1.75 | 2.17 | 2.68 | 3.27 | 3.95 | 4.71 | 5.42 |
| $T_p$ (s) | 8.68 | 8.54 | 8.14 | 7.65 | 7.51 | 7.63 | 8.04 | 8.58 | 9.16 | 9.58 |
| Probability | 0.087 | 0.213 | 0.266 | 0.230 | 0.134 | 0.052 | 0.013 | 0.002 | 0.001 | 0.001 |

## 2.4 Coordinate systems

This study employs two distinct coordinate systems. The global inertial coordinate system, illustrated in Fig. 7, offers a reference frame for examining how the ICR responds to varying environmental loading conditions and design parameters. By remaining unaffected by the structural displacements, it provides a clear view of how external forces influence the ICR. It is centred at the intersection of the mean waterline and the undisplaced tower axis (centroid of the undisplaced initial waterplane area). Conversely, the body-fixed coordinate system is used for design-focused evaluations, where the goal is to position the ICR relative to specific locations on the floating structure. This system initially coincides with the global inertial coordinate system but moves (both translates and rotates) with the floating body. Across the paper, the results are primarily presented in the global inertial frame. The local frame supports the interpretation of the observed design trends (distinction will be made where appropriate).

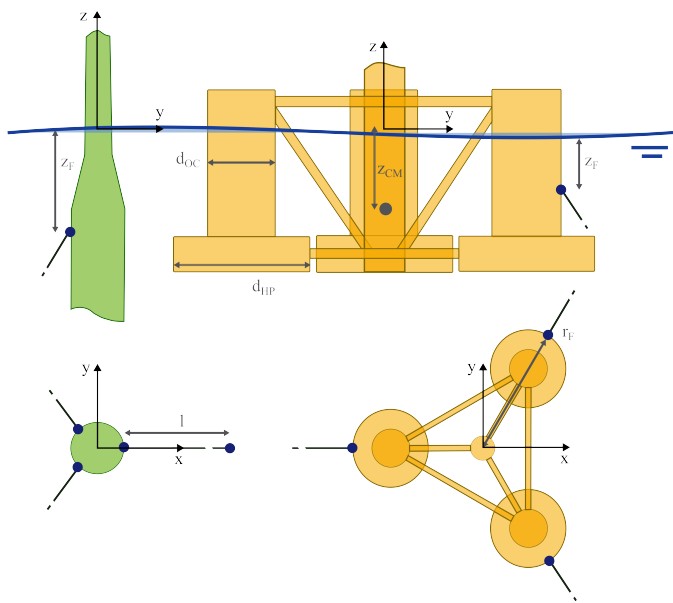

**Figure 7.** FOWT design variables. OC3 spar (left), OC4 semisubmersible (right).

## 2.5 Floating platform and turbine characteristics

The analysis is performed for two FOWT concepts: the OC3 Hywind spar (Jonkman, 2010) and OC4 semisubmersible (Robertson et al., 2014), both with the $5\,\mathrm{MW}$ reference rotor (Jonkman et al., 2009). The details of the reference designs are given in Tables 3–4 and Figure 7.

## 3 Environmental conditions effect

In a previous study (Patryniak et al., 2023), the ICR of FOWTs was only studied in regular waves, which provided a limited understanding of the behaviour in real operational scenarios. Therefore, this test aims to establish the dependence of the ICR on more complex environmental loading conditions, to understand the factors driving the ICR, and to establish the applicability of the ICR as a design metric.

## 3.1 Study setup

The sensitivity of the ICR to environmental loading conditions is studied through a series of simulations of gradually increasing complexity, as outlined in Table 5. All simulations are run for the case of the original design of the OC3 Hywind spar (Jonkman, 2010) with the $5\,\mathrm{MW}$ reference rotor (Jonkman et al., 2009) as the example system.

First, a simple test in regular waves is performed for a wide range of wave periods (or frequencies), keeping the wave height fixed (Group A in Table 5), Subsequently, the response in irregular waves is simulated, keeping the peak period and significant

**Table 3.** OC3 Hywind Spar and OC4 semisubmersible FOWT original designs characteristics. CG – centre of gravity; CB – centre of buoyancy.

| Property | Unit | OC3 spar | OC4 semisubmersible |
|---|---|---|---|
| **Platform, including ballast**: | | | |
| Draft | m | 120 | 20 |
| Diameter | m | 6.5–9.7 (tapered) | - |
| Offset column diameter | m | - | 12 |
| Base column diameter | m | - | 24 |
| Mass | kg | $7.47 \cdot 10^6$ | $1.35 \cdot 10^7$ |
| Vertical CG | m | $-89.92$ | $-13.46$ |
| Pitch inertia about CG | $\mathrm{kg\,m^2}$ | $4.23 \cdot 10^9$ | $6.83 \cdot 10^9$ |
| **Hydrostatics**: | | | |
| Vertical CB | m | $-61.17$ | $-13.17$ |
| Metacentric height | m | 28.7 | 10.7 |
| Stiffness coefficient: | | | |
| Heave | $\mathrm{N\,m^{-1}}$ | $3.33 \cdot 10^5$ | $3.84 \cdot 10^6$ |
| Pitch | $\mathrm{N\,m\,rad^{-1}}$ | $-4.99 \cdot 10^9$ | $-3.78 \cdot 10^8$ |
| **Infinite-frequency added mass**: | | | |
| Surge-surge | kg | $7.569 \cdot 10^6$ | $6.49 \cdot 10^6$ |
| Heave-heave | kg | $2.35 \cdot 10^5$ | $14.70 \cdot 10^6$ |
| Pitch-pitch | $\mathrm{kg\,m^2}$ | $3.7 \cdot 10^{10}$ | $7.21 \cdot 10^9$ |
| Surge-pitch | $\mathrm{kg\,m}$ | $-4.71 \cdot 10^8$ | $-8.51 \cdot 10^7$ |
| **Mooring system**: | | | |
| Fairlead height | m | $-70$ | $-14$ |
| Fairlead radius | m | 4.7 | 40.87 |
| Water depth (anchor position) | m | $-320$ | $-200$ |
| Anchor radius | m | 853.87 | 837.6 |
| Unstretched line length | m | 902.20 | 835.5 |
| Chain diameter (volume-equivalent) | m | 0.09 | 0.077 |
| Pretension | kN | 910 | 1100 |
| **Natural frequency**: | | | |
| Surge | Hz | 0.008 | 0.009 |
| Heave | Hz | 0.032 | 0.058 |
| Pitch | Hz | 0.034 | 0.037 |

wave height at the previously set values (Group B). Irregular sea state includes the component waves at various frequencies, with the most energy content around the peak frequency and contributions from lower- and higher-frequency waves. The complexity is then increased by including the effect of steady wind (from cut-in to cut-out), which puts the FOWT into a new equilibrium position (Group C), as well as the effect of the wind shear, which affects the vertical distribution of the wind load (Group D). Group E simulations are run in the rated wind speed condition, this time with turbulent wind inflow of varying

**Table 4.** NREL 5 MW reference rotor characteristics.

| Property | Unit | Value |
|---|---|---|
| RNA mass | kg | 350000 |
| Hub height | m | 90.0 |
| Cut-in, rated, cut-out wind speed | $\mathrm{m\,s^{-1}}$ | 3, 11.4, 25 |
| Rated wind speed thrust | kN | 805.11 |

**Table 5.** Impact of varied environmental loading on the ICR – conditions considered. $\gamma$ – peak shape factor.

| | Wind | | | | Waves | | | | Current |
|---|---|---|---|---|---|---|---|---|---|
| Group | Type | $V_s$ | $TI$ | Shear | Type | $H_s$ | $T_p$ | $\gamma$ | $V_c$ |
| (-) | (-) | $(\mathrm{m\,s^{-1}})$ | (-) | (-) | (-) | (m) | (s) | (-) | $(\mathrm{m\,s^{-1}})$ |
| A | - | - | - | - | Reg | 1.87 | 5–17.0 | - | - |
| B | - | - | - | - | JONSWAP | 1.87 | 7.47 | 1.0 | - |
| C | Steady | 3–25 | - | 0.2 | Reg | 1.87 | 7.47 | 1.0 | - |
| D | Steady | 11.4 | - | 0.1–0.4 | Reg | 1.87 | 7.47 | 1.0 | - |
| E | Turbulent | 11.4 | 0.1–0.3 | 0.2 | Reg | 1.87 | 7.47 | 1.0 | - |
| F | - | - | - | - | Reg | 1.87 | 7.47 | 1.0 | 0.5–1.2 |
| G | Turbulent | 11.4 | 0.17 | 0.2 | JONSWAP | 1.87 | 7.47 | 1.0 | 0.85 |

turbulence intensity. The turbulent wind is expected to increase the dynamic responses at very low frequencies not excited otherwise (the second-order wave loads are not modelled). Group F adds the effect of the subsurface current speed (the power low exponent is kept at the value $1/7$). Finally, Group G represents the condition likely encountered by an operational FOWT, including the combined effects of the stochastic wind, wave, and current loads.

The results obtained for each group will be presented in turn; all results are given in the global inertial coordinate system.

## 3.2 Results

The simulations conducted under Group A environmental conditions (regular waves) are analysed to understand how the external load period (or frequency) influences the motion behaviour and the ICR of the floating system. As presented in Fig. 8 and Table 6, the mean of the $x$-coordinate of ICR remains around zero regardless of the wave period, and the mean of the $z$-coordinate approaches zero as the wave period increases. As the wave period increases, the distribution of $z_{ICR}$ becomes wider (higher standard deviation).

Two mechanisms are important: i) Difference in the horizontal velocity's increase in two points on the structure due to increase in pitch motion as the wave period increases towards $31\,\mathrm{s}$, and ii) Higher horizontal velocity in both points due to increased surge motion as the wave period approaches $125\,\mathrm{s}$.

More specifically, the ICR is computed based on the intersection of the normals to the velocity vectors at two points on the structure located at $z_A = -120\,\mathrm{m}$ and $z_B = -12\,\mathrm{m}$. When the wave period approach the pitch natural period ($31\,\mathrm{s}$), the

pitch motion increases. Higher pitch motion about a point not coinciding with the midpoint between A and B induces a higher contribution to the horizontal velocity at the point further away from the centre of rotation. With higher pitch motion, the

imbalance between the two points becomes more pronounced, and the ICR shifts either higher or lower, depending on the direction of the pitch rotation. This leads to the ICR fluctuating over a wider range.

The second effect becomes more prominent when the wave period is closer to the surge natural period (125 s). In this case, higher surge motion imposes the same increase of the horizontal velocity in both points. Figure 9 illustrates that, for a given rotational velocity, if a significant horizontal translational velocity is superimposed, the ICR, defined as the point where normals

to the velocity vectors intersect, shifts either higher or lower depending on the direction of the translation direction and the instant of the periodic motion (i.e., the phase difference between the translation and rotation).[4] This results in both lower and higher values of the ICR, thereby widening the distributions as observed.

Additional figures supporting this discussion are provided in Figures B1– B2 in Appendix B.

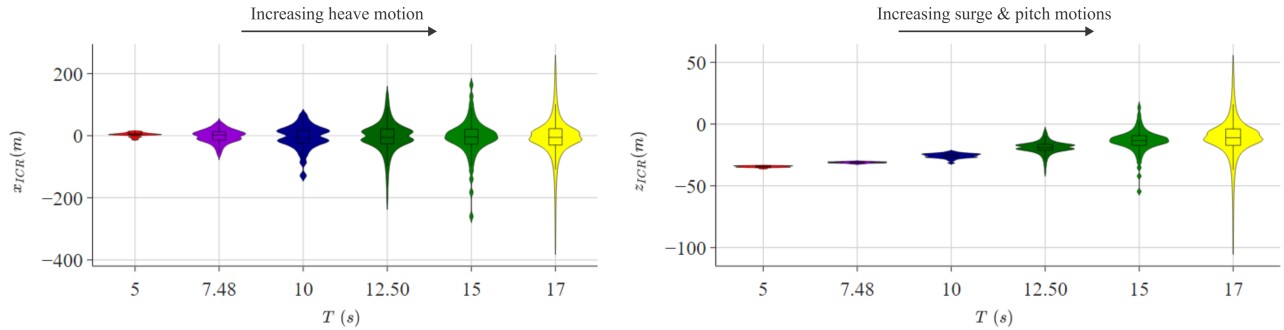

**Figure 8.** Effect of the regular wave period on the distribution of ICR coordinates (Group A).

**Table 6.** The ICR of a FOWT in regular waves (Group A).

|  | Surge | | Heave | | $x_{ICR}$ | | $z_{ICR}$ | |
|---|---|---|---|---|---|---|---|---|
| T | mean | std | mean | std | mean | std | mean | std |
| (s) | (m) | (m) | (m) | (m) | (m) | (m) | (m) | (m) |
| 5.00 | -0.09 | 0.07 | 0.02 | 0.00 | 3.55 | 6.28 | -34.52 | 0.63 |
| 7.47 | -0.10 | 0.21 | 0.02 | 0.02 | -1.70 | 21.49 | -31.12 | 0.61 |
| 10.00 | -0.09 | 0.38 | 0.02 | 0.05 | -6.38 | 39.50 | -25.79 | 1.94 |
| 12.50 | -0.09 | 0.51 | 0.02 | 0.09 | -4.77 | 51.61 | -19.08 | 5.15 |
| 15.00 | -0.09 | 0.62 | 0.03 | 0.14 | -6.69 | 62.63 | -13.90 | 10.08 |
| 17.50 | -0.09 | 0.76 | 0.02 | 0.16 | -6.85 | 70.59 | -11.43 | 17.74 |

The simulations of Group C environmental conditions (steady wind) are analysed to understand how the magnitude of the

non-fluctuating load applied at the rotor height influences the motion behaviour and the ICR of the floating system. The results

---

[4]Similarly, larger vertical velocity shifts the ICR horizontally away from the platform.

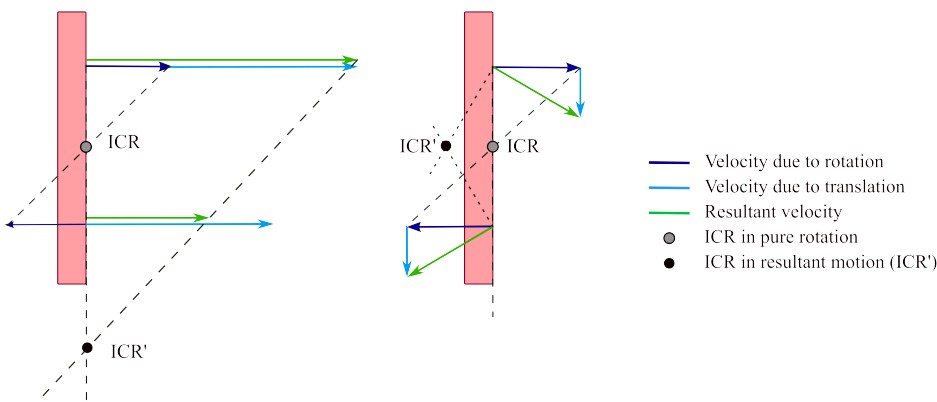

**Figure 9.** The effect of superposition of rotational and horizontal translational (left) and vertical translation (right) velocities on ICR position.

presented in Table 7 and Fig. 10 show that the mean of $x_{ICR}$ increases with wind speed, while the mean of $z_{ICR}$ follows a nonmonotonic trend. As seen previously for the regular wave cases, the distribution of the vertical coordinate gets wider as the wind speed increases.

**Table 7.** The ICR of a FOWT in steady wind (Group C).

| | Surge | | Heave | | $x_{ICR}$ | | $z_{ICR}$ | |
|---|---|---|---|---|---|---|---|---|
| $V_s$ | mean | std | mean | std | mean | std | mean | std |
| $(\mathrm{m\,s^{-1}})$ | (m) | (m) | (m) | (m) | (m) | (m) | (m) | (m) |
| 0.0 | -0.08 | 0.20 | 0.02 | 0.03 | -2.59 | 29.80 | -20.25 | 1.07 |
| 3.0 | 2.46 | 0.20 | -0.01 | 0.02 | -1.27 | 19.78 | -31.71 | 2.65 |
| 4.9 | 6.60 | 0.20 | -0.06 | 0.02 | -1.19 | 19.83 | -31.76 | 6.18 |
| 6.7 | 12.90 | 0.20 | -0.18 | 0.02 | -0.29 | 19.70 | -31.11 | 11.31 |
| 8.6 | 21.34 | 0.20 | -0.42 | 0.03 | 1.17 | 19.39 | -29.59 | 18.24 |
| 11.4 | 36.15 | 0.20 | -1.13 | 0.03 | 4.16 | 20.05 | -25.11 | 28.31 |

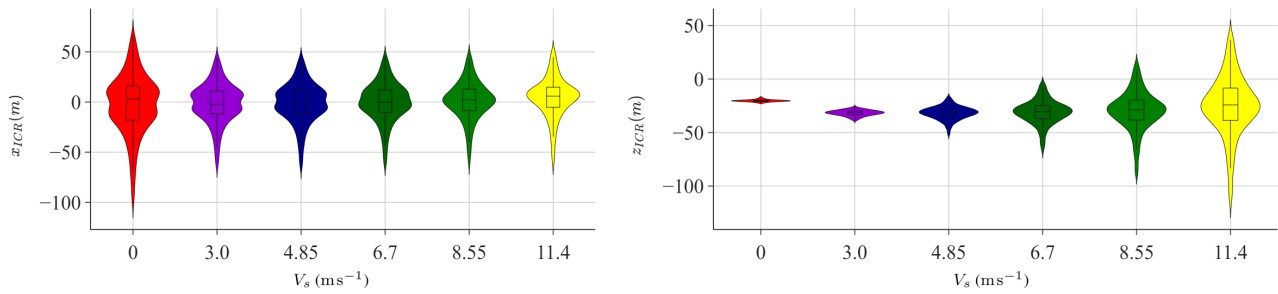

**Figure 10.** Effect of the steady wind speed on the distribution of ICR coordinates (Group C).

The next group of simulations (Group D) was conducted with a fixed mean hub-height wind speed of $11.4\,\text{ms}^{-1}$, varying the power exponent of the power law wind profile (shear). As demonstrated in Fig. 11, the distribution of $x_{ICR}$ remains mostly unaffected by the wind profile, as explained by negligible changes to the vertical motion (refer to Table 8). However, $z_{ICR}$ exhibits a strong dependence on the vertical distribution of wind. Notably, as the power exponent increases, the centre of application of the aerodynamic load (centre of effort) also shifts upward, resulting in a higher mean pitch angle. This change leads to two observable effects: i) an increase in pitch angle reduces the thrust coefficient, and ii) the rotor-averaged wind speed is altered. Depending on the trade-off between these two factors, the total aerodynamic thrust may be either lower or higher. This variation results in changes in the mean offset of the system, which subsequently affects the mooring loads and the horizontal velocities of the FOWT, which influence the behaviour of $z_{ICR}$.

**Table 8.** The ICR of a FOWT in steady wind with shear (Group D).

| | Surge | | Heave | | $x_{ICR}$ | | $z_{ICR}$ | |
|---|---|---|---|---|---|---|---|---|
| Shear | mean | std | mean | std | mean | std | mean | std |
| (-) | (m) | (m) | (m) | (m) | (m) | (m) | (m) | (m) |
| 0.10 | 2.46 | 0.20 | -0.01 | 0.02 | -1.27 | 19.78 | -31.71 | 2.65 |
| 0.15 | 12.90 | 0.20 | -0.18 | 0.02 | -0.29 | 19.70 | -31.11 | 11.31 |
| 0.20 | 36.15 | 0.20 | -1.13 | 0.03 | 4.16 | 20.05 | -25.11 | 28.31 |
| 0.25 | 6.60 | 0.20 | -0.06 | 0.02 | -1.19 | 19.83 | -31.76 | 6.18 |
| 0.30 | 21.34 | 0.20 | -0.42 | 0.03 | 1.17 | 19.39 | -29.59 | 18.24 |
| 0.35 | 35.93 | 0.20 | -1.11 | 0.03 | 4.27 | 20.01 | -25.59 | 28.96 |
| 0.40 | 35.95 | 0.20 | -1.11 | 0.03 | 4.07 | 20.16 | -25.49 | 28.88 |

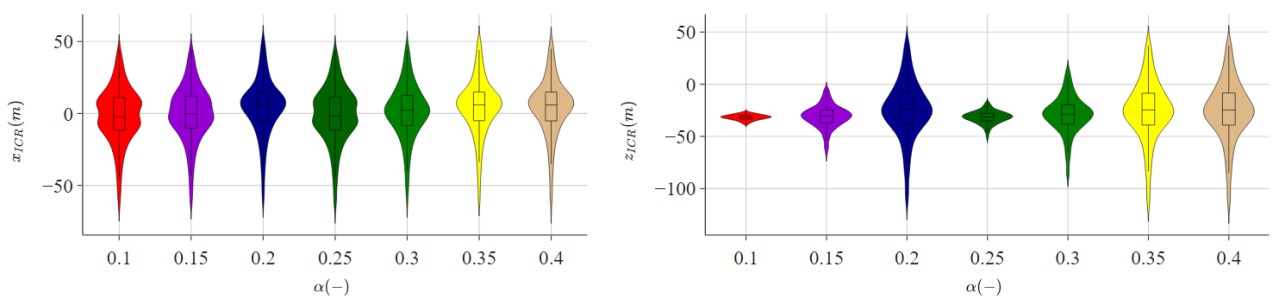

**Figure 11.** Effect of the wind shear on the distribution of ICR coordinates (Group D).

Figures 12–13 and Tables 9–10 demonstrate that the ICR distribution does not significantly change in response to changing turbulence intensity (Group E) or subsurface current velocity (Group F).[5] The distributions at all turbulence levels remain normal, with a slightly increasing standard deviation due to increased horizontal load (higher turbulence intensity leads to

---

[5]Notice the small range on the y-axis.

higher aerodynamic fluctuating load for any mean wind speed given). The current load on the slender cylinder of the spar floating structure is insignificant compared to other loads on the system.

**Table 9.** The ICR of a FOWT in turbulent wind (Group E).

| TI (%) | Surge mean (m) | Surge std (m) | Heave mean (m) | Heave std (m) | $x_{ICR}$ mean (m) | $x_{ICR}$ std (m) | $z_{ICR}$ mean (m) | $z_{ICR}$ std (m) |
|---|---|---|---|---|---|---|---|---|
| 0 | 36.15 | 0.20 | -1.13 | 0.03 | 4.16 | 20.05 | -25.11 | 28.31 |
| 10 | 36.15 | 0.25 | -1.13 | 0.04 | 4.11 | 20.26 | -25.15 | 29.67 |
| 15 | 36.15 | 0.30 | -1.13 | 0.04 | 3.77 | 20.33 | -24.81 | 31.26 |
| 20 | 36.24 | 0.35 | -1.14 | 0.05 | 3.84 | 20.92 | -25.09 | 33.20 |
| 25 | 36.38 | 0.39 | -1.15 | 0.05 | 3.93 | 21.37 | -25.40 | 34.73 |
| 30 | 36.57 | 0.43 | -1.16 | 0.06 | 3.63 | 22.13 | -26.39 | 37.58 |

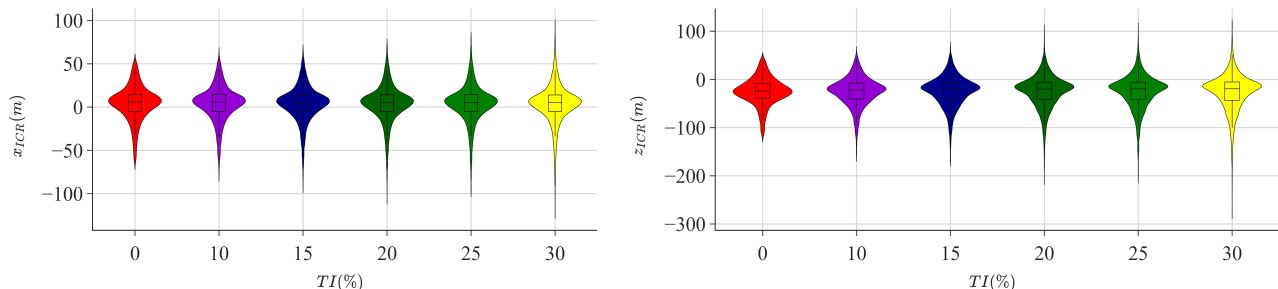

**Figure 12.** Effect of the turbulence intensity on the distribution of ICR coordinates (Group E).

**Table 10.** The ICR of a FOWT simulated in varying current velocities (Group F).

| $V_c$ (m s$^{-1}$) | Surge mean (m) | Surge std (m) | Heave mean (m) | Heave std (m) | $x_{ICR}$ mean (m) | $x_{ICR}$ std (m) | $z_{ICR}$ mean (m) | $z_{ICR}$ std (m) |
|---|---|---|---|---|---|---|---|---|
| 0.00 | -0.10 | 0.21 | 0.02 | 0.02 | -1.55 | 21.39 | -31.10 | 0.62 |
| 0.25 | 0.39 | 0.17 | 0.02 | 0.02 | -1.36 | 19.34 | -31.09 | 0.60 |
| 0.50 | 2.01 | 0.20 | 0.02 | 0.02 | 0.01 | 20.93 | -31.17 | 0.66 |
| 0.70 | 3.94 | 0.20 | 0.01 | 0.02 | -0.19 | 21.35 | -31.18 | 0.82 |
| 0.80 | 5.27 | 0.20 | 0.01 | 0.02 | -0.26 | 20.60 | -31.23 | 0.91 |
| 1.00 | 8.41 | 0.21 | -0.02 | 0.02 | -0.35 | 23.20 | -31.09 | 1.23 |
| 1.20 | 12.39 | 0.21 | -0.07 | 0.02 | -0.01 | 23.24 | -31.16 | 1.60 |

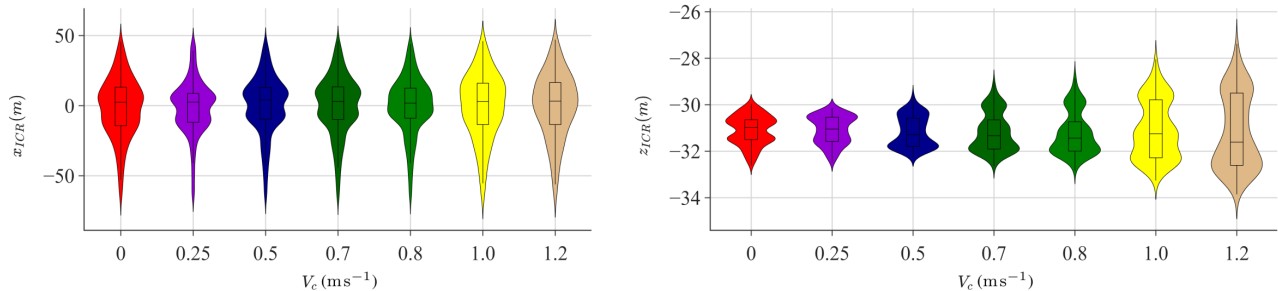

**Figure 13.** Effect of the current speed on the distribution of ICR coordinates (Group F).

Finally, Fig. 14 compares the distributions of the ICR coordinates of a FOWT placed in increasingly complex environmental conditions. While $x_{ICR}$ appears consistent across all cases, the complexity of the loading significantly influences $z_{ICR}$,

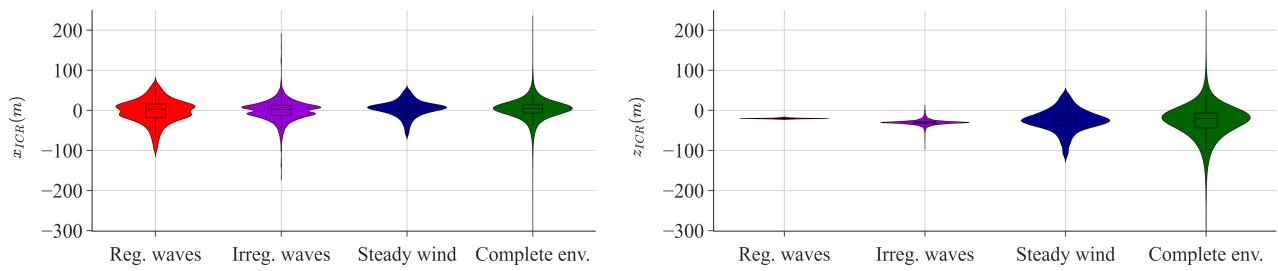

**Figure 14.** Effect of the increased environmental loading complexity on the distribution of ICR coordinates. The regular/irregular wave cases only include regular/irregular wave loads, the steady wind case includes steady wind in addition to irregular waves, and the complete environment includes irregular waves, turbulent wind, and current loads.

particularly in terms of its standard deviation. In the presence of regular waves, the vertical coordinate of ICR is confined to a narrow range. However, as irregular waves, wind, and current are gradually introduced, this range expands considerably.

Figure 15 provides a more detailed view of the effect of the environment complexity on $z_{ICR}$ range. For each loading condition case, the confidence interval bounds were calculated using the mean and standard deviation of $z_{ICR}$ for various confidence levels. The ranges displayed in the figure represent the differences between the upper and lower bounds, indicating
the intervals within which the true mean $z_{ICR}$ is expected to lie with the specified confidence level. For any confidence level, more complex environmental loading leads to a broader range. This suggests that the complexity of the loading significantly impacts the variability of $z_{ICR}$ and the potential for extreme values. Importantly, broader ranges are also related to increased uncertainty in predicting the true mean of ICR, which may affect the usefulness of the ICR as a metric in the FOWT design process, as it is less likely to remain near a target location.
As reported in Table 11, the cases with no stochastic loads (i.e., regular wind and/or steady wind) show a relatively low K-S statistic and a high p-value (these metrics are introduced in Sect. 2.2), meaning that the ICR closely follows a normal distribution in these scenarios. In contrast, the low p-value observed for the stochastic loading cases (irregular waves and/or

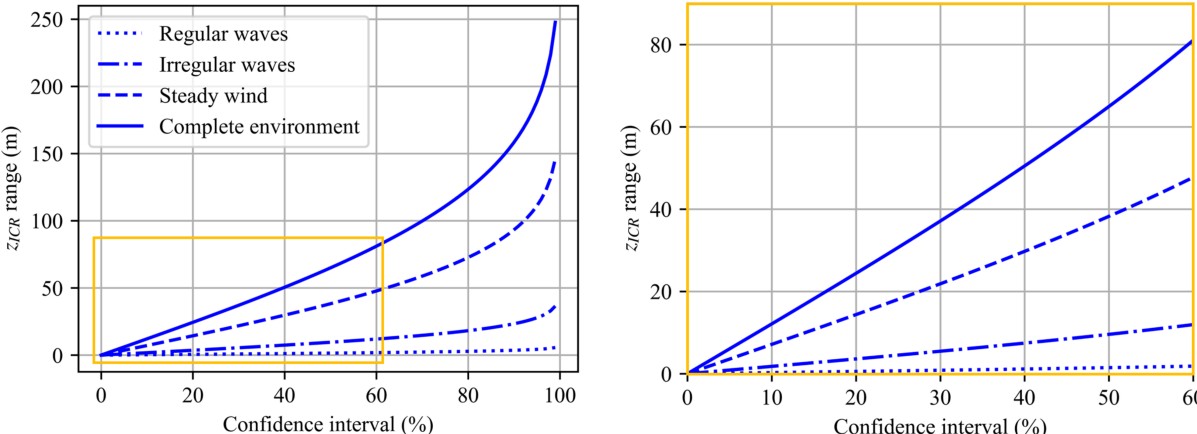

**Figure 15.** $z_{ICR}$ range for different confidence intervals.

turbulent wind) suggests that we cannot reach the same conclusion regarding the normality of the ICR distribution in those instances. In those cases, the observed distribution of $z_{ICR}$ has significantly increased skewness (more asymmetric) and excess

**Table 11.** The ICR of a FOWT under environmental loading conditions of varying complexity. Each row represents one case of a group listed in Table 5.

| | Surge | | Heave | | $x_{ICR}$ | | | | $z_{ICR}$ | | | |
| Condition | mean | std | mean | std | mean | std | K-S | p | mean | std | K-S | p |
| | (m) | (m) | (m) | (m) | (m) | (m) | (-) | (-) | (m) | (m) | (-) | (-) |
| Reg. waves (A) | -0.08 | 0.20 | 0.02 | 0.03 | -2.59 | 29.80 | 0.11 | 0.58 | -20.25 | 1.07 | 0.14 | 0.28 |
| Irreg. waves (B) | -0.08 | 0.13 | 0.02 | 0.02 | -1.04 | 25.52 | 0.10 | 0.72 | -30.10 | 7.09 | 0.16 | 0.16 |
| Steady wind (C) | 36.15 | 0.20 | -1.13 | 0.03 | 4.16 | 20.05 | 0.07 | 0.98 | -25.11 | 28.31 | 0.12 | 0.46 |
| Complete env. (G) | 40.48 | 0.24 | -1.38 | 0.03 | 2.28 | 29.23 | 0.13 | 0.34 | -26.87 | 47.36 | 0.15 | 0.17 |

kurtosis (heavier tails and sharper peaks), as illustrated in Fig. 16.

## 4  Design variables sensitivity

For any environmental loading condition, the motion response of different FOWT designs differs due to unique hydrostatic, hydrodynamic, mooring, and inertial characteristics.[6] Therefore, this test investigates which design features (or variables) can be adjusted to achieve the desired ICR "location".

Two floating systems are considered in turn: a spar and a semisubmersible. The study includes four design variables common for both concepts: mooring line length, fairlead vertical position and radius, and platform centre of mass vertical position.

---

[6]The impact of design parameters on the system dynamics were studied in (Jonkman et al., 2022)

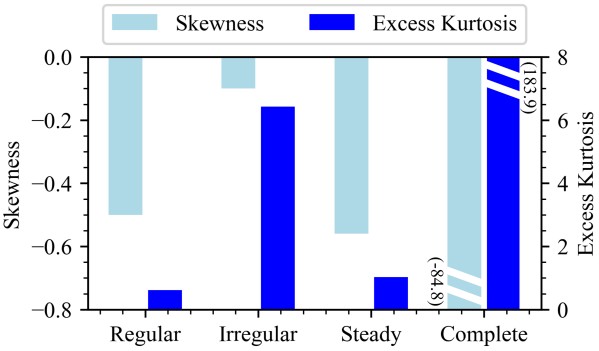

**Figure 16.** Skewness and kurtosis of $z_{ICR}$ distribution for different environmental loading conditions.

Additionally, two variables are specific to the semisubmersible platform (offset column and heave plate diameters). These are illustrated in Fig. 7 and listed in Table 12. The anchor vertical position (water depth) is adjusted together with the fairlead vertical position variable to keep constant anchor-fairlead distance and only vary the point of application of the mooring load on the platform.

Note that this set of design variables is not exhaustive, and each parameter is varied independently. In particular, the fairlead position is adjusted without consideration of the floating platform dimensions, which keeps the effects isolated but neglects the proper physical interface between the fairleads and the platform. The catenary mooring line length is modified without altering the fairlead and anchor positions, resulting in changes to the line angle at the fairlead, the suspended line profile, and the portion of the line resting on the seabed. Likewise, the ratio between the offset column and heave plate diameters is not maintained when either variable is adjusted.

The broad range of design variables ensures that the study covers the entire design space, including some unconventional but realistic designs demonstrating a wide range of motion responses. Special attention was given to the mooring line length parameter, with the range chosen based on mooring pretension. Specifically, the study examined mooring line lengths that resulted in a $\pm 40\%$ variation in pretension compared to the original design, to cover a plausible range of designs.

Note that these design variable perturbations may alter the mean heave (draft) of the floating platform due to changes in displacement volume (buoyancy) and/or the weight of the structure. This effect could significantly impact the response of the FOWT and may be undesirable, as the draft is a critical design parameter established early in the design process. It is often constrained by factors such as port depth and manufacturing or assembly capabilities. Therefore, an additional step was implemented whenever the underwater geometry was modified: the floating platform was re-ballasted to maintain the original draft. This not only affected the total weight of the system but also influenced the weight distribution and center of mass.

For each concept, the design variables were perturbed one at a time, and the change in the ICR statistics (in the global inertial coordinate system) was studied, as illustrated in Fig. 17. As introduced in Section 2.3, the results for each design variant were averaged over 10 environmental condition bins, with weights based on bin probability. The design variables presented in the

**Table 12.** Original values and ranges of design variables. All values in (m).

| | OC3 spar | | | OC4 semi | | |
|---|---|---|---|---|---|---|
| | Original | Lower Bound | Upper Bound | Original | Lower Bound | Upper Bound |
| Line length ($l$) | 902.2 | 891.6 | 924.1 | 835.5 | 828.5 | 850.0 |
| Fairlead radius ($r_F$) | 5.2 | 4.0 | 8.0 | 40.87 | 20.0 | 50.0 |
| Fairlead z ($z_F$) | -70.0 | -120 | 3.5 | -14.0 | -20.0 | 1.0 |
| Platform CM ($z_{CM}$) | -89.92 | -110 | -84.29 | -8.66 | -10.0 | -7.0 |
| Offset column D ($d_{OC}$) | - | - | - | 12.0 | 10.80 | 13.20 |
| Heave plate D ($d_{HP}$) | - | - | - | 24.0 | 21.60 | 26.40 |

figure were normalised to 0–1 for clarity and unbiased comparison between the variables of different magnitudes. The absolute ranges of the variables are given in Table 12 for reference.

Based on the analysis of the spar platform in Fig. 17, it is clear that variations in certain design variables significantly impact the position of the ICR. In particular, the results highlighted the significant influence of the centre of mass location on the ICR statistics. This effect was most evident for the $z$-coordinate (lower two plots), where the mean and standard deviation of the ICR showed strong negative and positive trends with the vertical position of the centre of mass, respectively. The vertical position of the fairlead was also found to be an important feature affecting the mean of $x_{ICR}$. The effects of other variables on the ICR metrics were relatively minor. Notably, no clear trends were observed for the standard deviation of $x_{ICR}$, suggesting that the variables examined do not significantly influence this metric independently.

For the semisubmersible-type FOWT, Fig. 18 demonstrates that the design variables have a more limited impact on the mean of ICR than was the case for the spar platform. Specifically, the mean position of the ICR remained within approximately $4\,\mathrm{m}$ across all design variations, indicating a smaller impact from design changes. Higher variability was observed in the standard deviation of the ICR. Notably, the offset column and heave plate diameters emerged as significant parameters, affecting both the mean and the fluctuations in the ICR position. The influence of other design variables on the ICR was relatively minor.

To provide more quantitative insights, a sensitivity analysis was conducted using the data from Figs. 17 and 18 to compute the Spearman correlation coefficient for each pair of design variables and ICR statistics (details in Sect. 2.2). As demonstrated in Figs. 19 and 20, the sensitivity analysis confirmed the earlier observations. Additionally, for the case of the semisubmersible platform, it showed a non-negligible relationship between the mean $x$-coordinate of the ICR and the mooring line design parameters: length, fairlead height, and fairlead radius (correlation coefficient of about $\pm 0.3$).

These quantitative results allow for ranking the design variables according to their impact on the ICR statistics, helping to identify which variables offer the greatest potential for ICR-driven design optimisation. This will be further explored in Sect. 6.

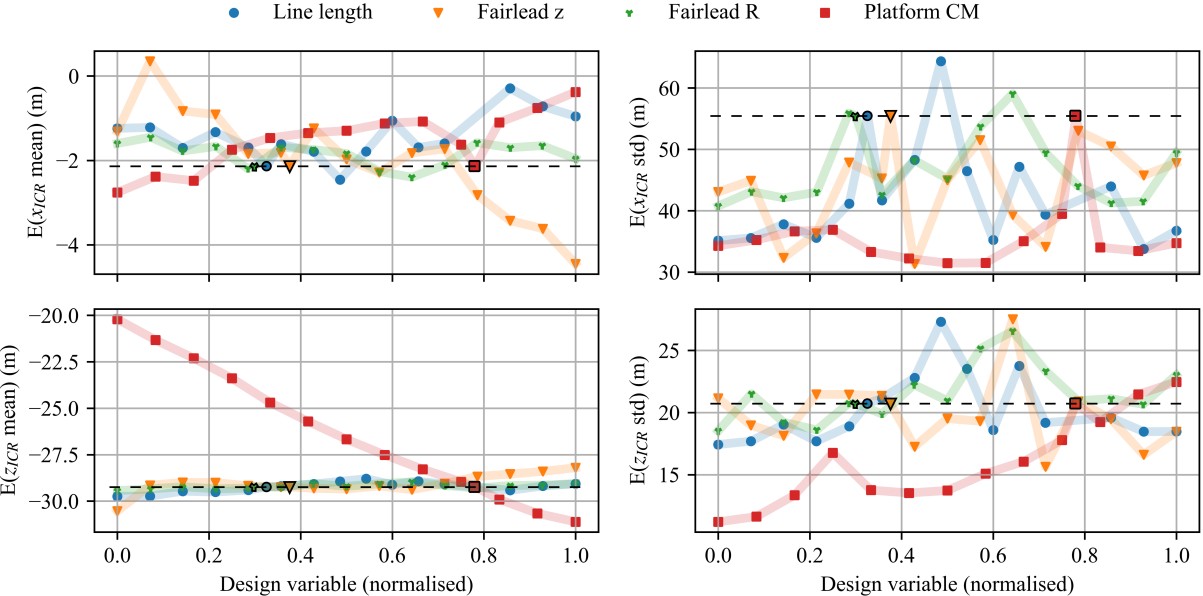

**Figure 17.** ICR statistics for different one-at-a-time design variable perturbations for a spar FOWT. Original design values marked with black outline. Results averaged over 10 environmental conditions.

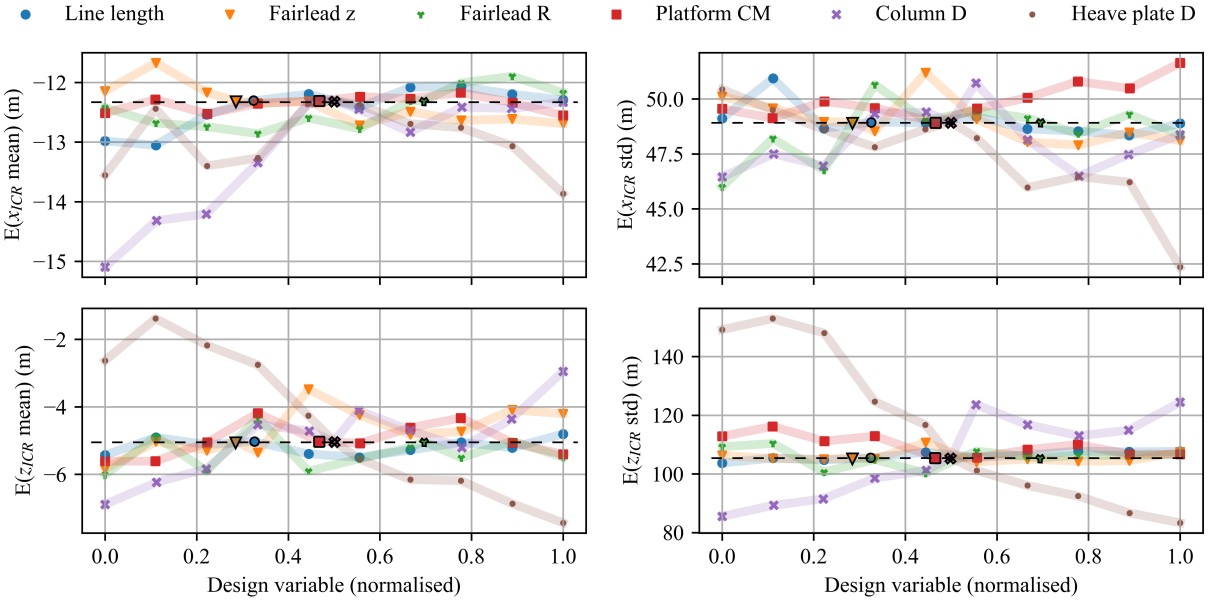

**Figure 18.** ICR statistics for different one-at-a-time design variable perturbations for a semisubmersible FOWT. Original design values marked with black outline. Results averaged over 10 environmental conditions.

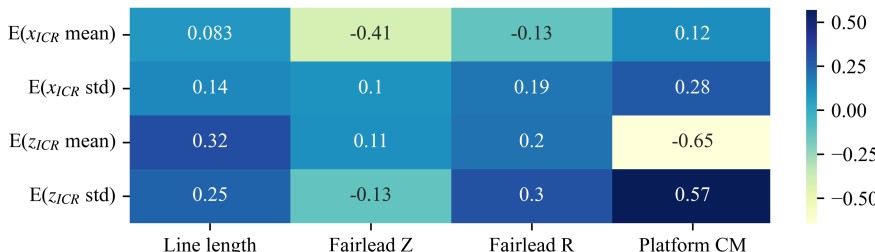

**Figure 19.** Correlation between the statistics of ICR and the design variables for the spar-type FOWT. See Fig. C1 in Appendix C for correlations of all responses.

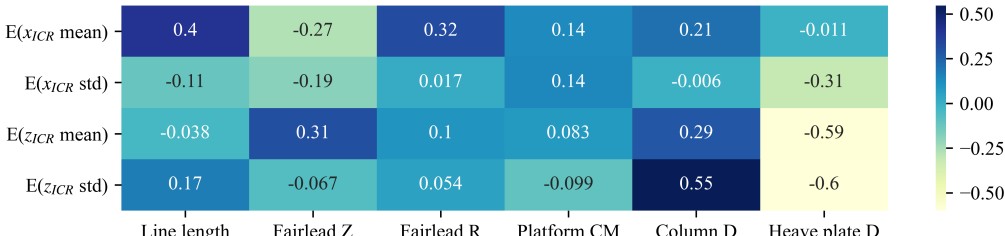

**Figure 20.** Correlation between the statistics of ICR and the design variables for the semisubmersible-type FOWT. See Fig. C2 in Appendix C for correlations of all responses.

## 5  Responses sensitivity

To determine which responses of the spar FOWT can be improved by incorporating ICR in the design process, a sensitivity analysis was performed based on the full factorial design of experiments of the two most impactful design variables identified in Sect. 4 (vertical fairlead position and platform centre of mass), resulting in $64$ variations of the floating systems.

Each design variation was simulated in 10 environmental conditions based on the binned metocean data obtained for the Scottish NE8 site, and the results plotted refer to the average results weighted with the probability of occurrence of a given environmental condition, as explained in Sect. 2.3. The metrics considered included the nacelle fore-aft acceleration (referred to as NcIMUTAxs in tables and figures), blade root out-of-plane and tower base fore-aft bending moments (RootMyc1, TwrB-sMyt), and the most loaded mooring line tension at the fairlead (FAIRTEN2), as well as the computed $x$ and $z$ coordinates of ICR (in the global inertial coordinate system). For each pair of responses, distribution statistics and the Spearman rank correlation coefficients were computed, as detailed in Sect. 4.

As illustrated in the correlation matrix in Fig. 21, a significant negative correlation ($-0.73$ to $-0.59$) was observed between the standard deviation of the ICR horizontal coordinate and several responses, including nacelle acceleration and both the mean and standard deviation of tower base and blade root bending moments. A strong negative correlation was also observed between the mooring tension and the mean of $x_{ICR}$ ($-0.61$ to $-0.59$). A weaker yet significant correlation ($0.43$ to $0.46$) exists between the mean of $x_{ICR}$ and the mean of the tower base and blade root loads. The standard deviation of $z_{ICR}$ also

exhibits a moderate correlation with these loads (0.41 to 0.45). Other statistics, such as skewness and kurtosis, were computed and analysed; however, they did not yield significant correlations and are therefore omitted for clarity. Similarly, the expected value is not included, as it was found to be very close to the mean of the distributions.

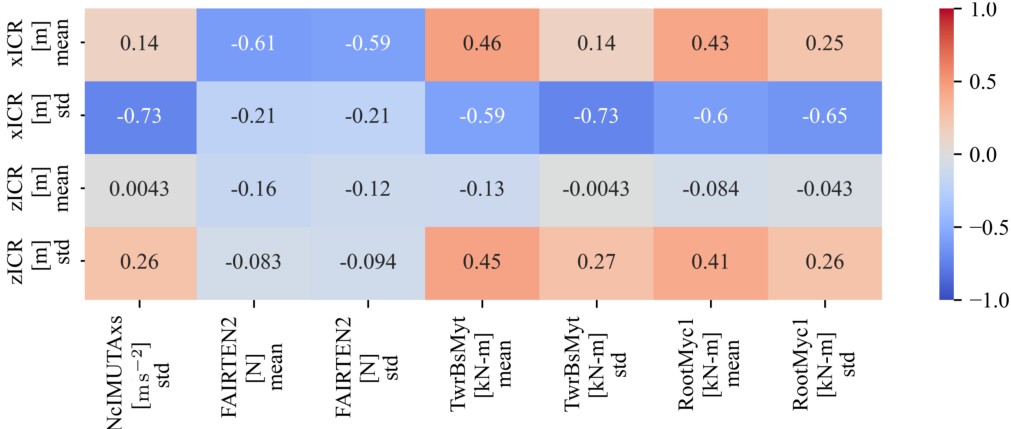

**Figure 21.** Spearman correlation between the statistics of ICR and the loads, obtained by varying the spar design features.

The same procedure was followed for the semisubmersible type of FOWT, investigating the responses of 150 designs of varying offset column diameter, heave plate diameter, and line length according to the full factorial sampling strategy. As seen in Fig. 22, the strongest correlation was found between the mean of $z_{ICR}$ and all the responses considered (up to $-0.83$). A weaker correlation around $\rho = -0.3$ was found between all statistics of ICR and two responses: nacelle acceleration and the standard deviation of the tower base bending moment.

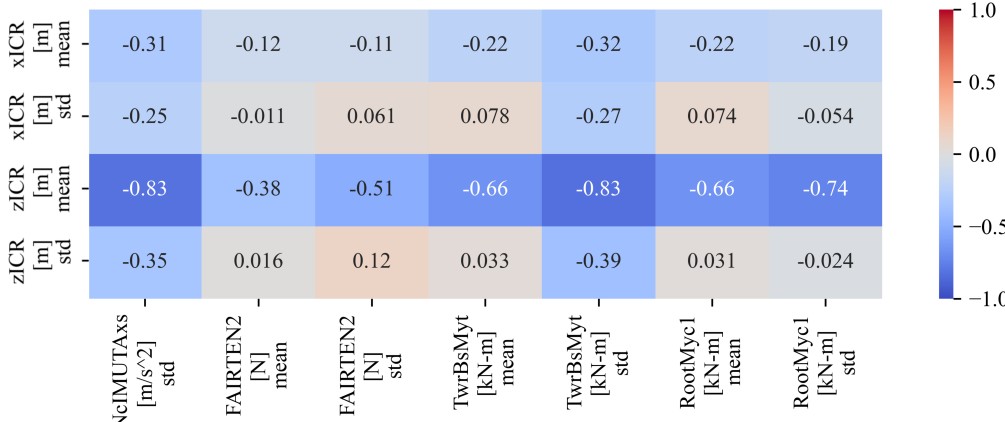

**Figure 22.** Spearman correlation between the statistics of ICR and the loads, obtained by varying the semisubmersible design features.

The presented correlation analysis provides information about the strength and direction of a monotonic relationship. However, it does not give specific information about the magnitude of the change of one parameter in response to a shift in another.

Therefore, to assess the potential impact of the ICR on the responses of interest, the correlation results should be interpreted together with the information about the ranges of the responses' statistics, as provided in Table 13. By analysing the minimum, median, and maximum values, it can be concluded that the ranges of the responses highly correlated with ICR are sufficiently broad to be regarded as useful design metrics, offering adequate design flexibility. For example, the strong correlation between the mean of $z_{ICR}$ and the standard deviation of the tower base bending moment, shown in Fig. 22, is significant because both variables span meaningful ranges: the range of $z_{ICR}$ mean from $-35.47\,\mathrm{m}$ to $-18.86\,\mathrm{m}$ corresponds to a tower base bending moment standard deviation range from $3.33\,\mathrm{MN\,m}$ to $6.48\,\mathrm{MN\,m}$.

**Table 13.** Summary statistics of responses for spar and semisubmersible structures based on data used for the correlation analysis.

| | | Spar | | | Semisubmersible | | |
|---|---|---|---|---|---|---|---|
| | Unit | Min | Median | Max | Min | Median | Max |
| NcIMUTAxs std | $\mathrm{m\,s^{-2}}$ | 0.06 | 0.08 | 0.12 | 0.05 | 0.08 | 0.14 |
| FAIRTEN2 mean | kN | 637.04 | 1276.43 | 2977.12 | 865.86 | 1409.13 | 2190.55 |
| FAIRTEN2 std | kN | 0.59 | 7.80 | 29.98 | 5.83 | 17.94 | 61.21 |
| TwrBsMyt mean | MN m | -5.58 | 24.14 | 100.74 | -0.37 | 23.61 | 73.89 |
| TwrBsMyt std | MN m | 3.33 | 4.35 | 6.48 | 1.85 | 2.99 | 5.16 |
| RootMyc1 mean | MN m | 0.80 | 3.31 | 9.50 | 0.86 | 3.91 | 9.47 |
| RootMyc1 std | MN m | 0.08 | 0.19 | 0.76 | 0.07 | 0.22 | 0.68 |
| xICR mean | m | -8.43 | -1.88 | 4.72 | -19.22 | -13.00 | -5.98 |
| xICR std | m | 20.40 | 32.44 | 110.56 | 38.28 | 47.71 | 102.22 |
| zICR mean | m | -35.47 | -26.60 | -18.86 | -18.26 | -5.14 | 9.61 |
| zICR std | m | 5.97 | 13.84 | 46.28 | 68.15 | 106.96 | 337.67 |

## 6 Design case studies

The insights gained in the previous sections can be applied in design scenarios. The case studies presented here are defined based on the identification of the responses most sensitive to the ICR (i.e., which can be improved by adjusting ICR) (Sect. 5), and the design variables that have a significant impact on ICR and enable these improvements (Sect. 4). The first case study is done for the case of the spar-supported FOWT, and the second case focuses on the semisubmersible-type FOWT. The correlations used in the case studies are summarised in Fig. 23 and detailed in the subsequent sections.

### 6.1 Spar sizing design case study

Given the observations made in Sect. 5 based on Fig. 21, one major area would be particularly worth investigating in the context of design improvements through appropriate ICR "location". Namely, by maximising the standard deviation of the horizontal coordinate of ICR (i.e., by shifting it towards the positive $x$ direction), one could reduce the nacelle acceleration and the tower

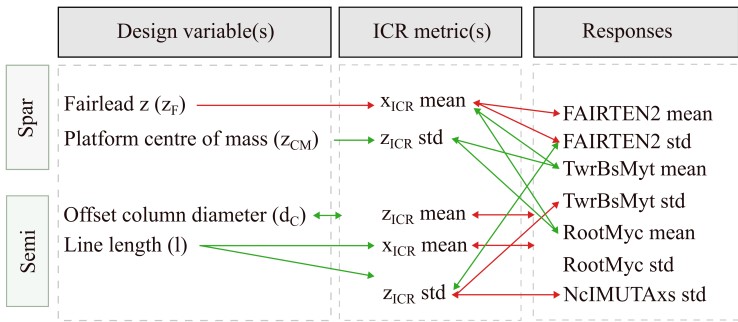

**Figure 23.** Case studies: design variables considered and their correlation with the ICR and responses statistics. Green/red arrows indicate a positive/negative correlation.

base and blade root bending moments[7]. Lower nacelle acceleration is desired from the point of view of the wind turbine's efficient and safe operation; reducing the standard deviation of the tower and blade loads is important for fatigue damage reduction. However, for these improvements to be possible, it would be necessary to find a set of design variables correlated with the standard deviation of the horizontal coordinate of the ICR and the mean of the ICR coordinates. As can be seen in the correlation matrix in Fig. 19 in Sect. 4, none of the variables considered gives any control over the standard deviation of $x_{ICR}$. Therefore, this metric cannot be modified in a design scenario.

Instead, the mean of $x_{ICR}$ was seen to be strongly negatively correlated with the mean and standard deviation of the mooring loads and, at the same time, negatively correlated with the fairlead position. Therefore, it seems desirable to lower the fairlead position to increase the mean of $x_{ICR}$ to, in turn, decrease the mean and dynamic tension. However, the mean of $x_{ICR}$ is also positively correlated with the mean of the tower and blade loads; therefore, shifting the ICR towards a position which improves the tension characteristics might increase the tower and the blade loads. This effect can be compensated for by simultaneously lowering the platform centre of mass to effectively reduce the tower and blade loads. Therefore, this case study will attempt to improve the mooring loads by varying two design variables: the vertical position of the fairlead and the platform centre of mass, keeping the tower base and blade roots mostly unaffected.

Figure 24 illustrates the design space in the form of $2D$ trends for the mean and standard deviation of the ICR coordinates and responses identified as correlated, for all combinations of the two selected design variables (full factorial design). While the sensitivity of the mean $x_{ICR}$ to each variable is largely independent of the other variable (top left subplot), more interdependence is seen in the case of the standard deviation of $z_{ICR}$, where the rate of ICR increase with the centre of mass depends on the fairlead position and vice versa. Regarding the loads, the mooring line tension at the fairlead is significantly more sensitive to variations in fairlead position than it is to the centre of mass. The impact of the two design variables on the tower base and blade root bending moments is more balanced, with both variables affecting the loads significantly.

As previously outlined in the diagram of Fig. 23 and additionally illustrated in Fig. 24, the design modifications follow the steps listed below:

---

[7]Note that the nacelle acceleration is one of the factors contributing to the tower base bending moment, therefore these responses are not independent.

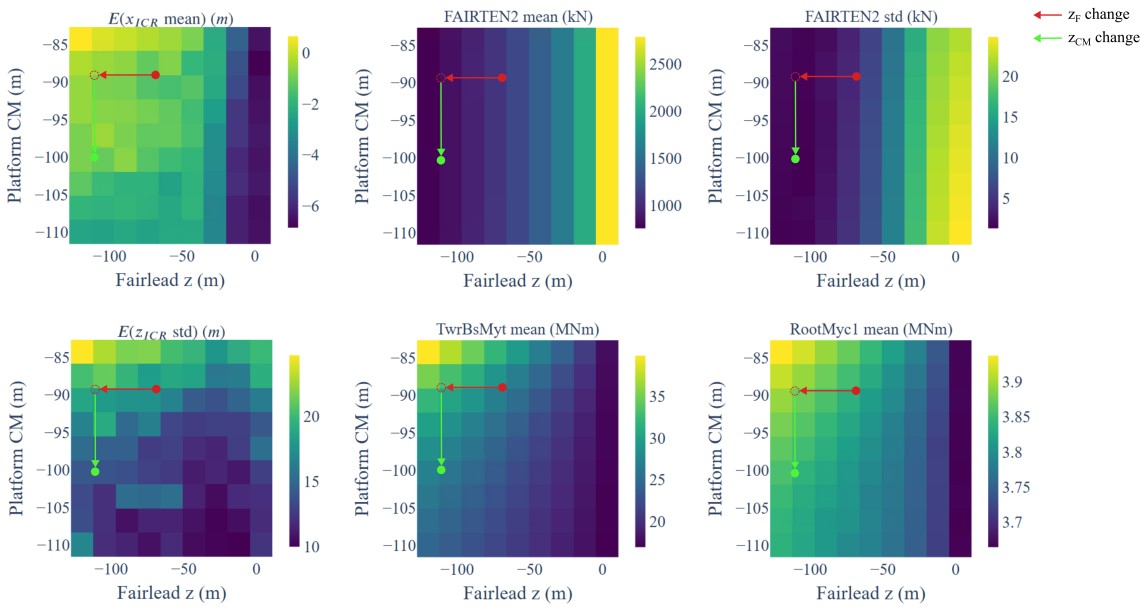

**Figure 24.** Design space visualisation: ICR and response statistics as functions of two selected design variables for the spar-type FOWT.

– Lower the fairlead position to increase the mean $x_{ICR}$ (top-left plot, red arrow).

– This leads to a reduction of the mean and dynamic mooring tension (top-middle and top-right plots, red arrows) and to an unfavourable increase in tower base and blade root bending moments (bottom-middle and bottom-right plots, red arrows).

– Lower the platform centre of mass to lower the standard deviation of $z_{ICR}$ (bottom-left plot, green arrow).

– This brings the tower base and blade root loads back close to the original values while retaining the reduced mooring tension.

The extent of the variable adjustments is limited by practical considerations: The fairlead may not be practical to be constructed at the very bottom of the spar structure, and the centre of mass of the platform is limited by the ballast placement.[8]

OpenFAST results of the original and the modified designs are compared in Table 14 and Fig. 25. As expected, the most significant change was seen in the mooring loads, with $26\%$ and $69\%$ reductions in the mean and dynamic tension, respectively, improving both the ultimate strength and fatigue characteristics of the design. These changes were accompanied by shifting the mean of the body-fixed coordinate system ICR closer to the fairleads position. The softer mooring system resulted in vastly increased mean surge offset of $21.73\,\mathrm{m}$. Although this was a $50\%$ increase over the original design, the value is still reasonable ($7\%$ of water depth assumed in this study). Importantly, the reduction of mooring loads was achieved without significantly

---

[8]Implementing such adjustments may necessitate additional design modifications—for example, lowering the center of mass could require a deeper draft structure. While some consideration has been given to the practicality of these design variations, their overall feasibility is not guaranteed.

compromising the other structural loads: the only response negatively affected was the tower base bending moment, which increased by less than 5 %.

**Table 14.** Comparison of the original and the ICR-informed adjusted spar FOWT designs' responses. Body-fixed frame ICR values in brackets.

| Metric | Unit | Mean | | Std | |
|---|---|---|---|---|---|
| | | Original | Adjusted | Original | Adjusted |
| $x_{ICR}$ | m | -2.13 (21.60) | -0.79 (12.63) | 54.79 (33.35) | 33.40 (54.80) |
| $z_{ICR}$ | m | -28.69 (-23.27) | -24.31 (-28.86) | 19.83 (15.36) | 15.30 (19.58) |
| FAIRTEN2 | N | 1.09E+06 | 8.06E+05 | 4.24E+03 | 1.31E+03 |
| TwrBsMyt | kN m | 2.68E+04 | 2.64E+04 | 4.04E+03 | 4.21E+03 |
| RootMyc1 | kN m | 3.83E+03 | 3.85E+03 | 2.37E+02 | 2.32E+02 |
| Surge | m | 13.93 | 21.73 | 0.12 | 0.13 |
| Heave | m | -0.16 | 1.04 | 0.02 | 0.02 |
| Pitch | ° | 1.62 | 1.53 | 0.06 | 0.06 |
| NcIMUTAxs | $m s^{-2}$ | | | 0.11 | 0.11 |

**Table 15.** The original and the ICR-informed adjusted spar FOWT designs features.

| Feature | Unit | Original | Adjusted |
|---|---|---|---|
| Fairlead $z$ | m | -70.00 | -110.00 |
| Platform $z_{CM}$ | m | -89.92 | -100.00 |

## 6.2 Semisubmersible sizing design case study

The second case study follows the procedure presented for the spar-type FOWT case, but this time, it is performed for the semisubmersible-supported floating system.

Based on the observations made in Sect. 5, the most promising design task would be to adjust the structure's features to increase the mean of $z_{ICR}$, to decrease all loads simultaneously. Defining a single design metric (the mean of $z_{ICR}$) that accounts for multiple metrics (nacelle acceleration, mooring, tower, and blade root loads) might be beneficial in design optimisation scenarios, as it reduces the size of the optimisation problem, potentially leading to a less time-consuming and more robust optimisation process.

By examining the correlation matrix in Fig. 20, it can be seen that the mean of $z_{ICR}$ is strongly correlated with the heave plate diameter ($\rho = -0.59$) and weakly correlated with the fairlead height ($\rho = 0.31$) and offset column diameter ($\rho = 0.29$). However, a change in one variable can lead to undesirable changes in multiple ICR statistics. Here, reducing the heave plate diameter to increase the mean of $z_{ICR}$ would also lead to an increase in the standard deviation of $z_{ICR}$, which would lead to an increase in the dynamic mooring loads. Fairlead height is also negatively correlated with the mean of $x_{ICR}$ which is positively

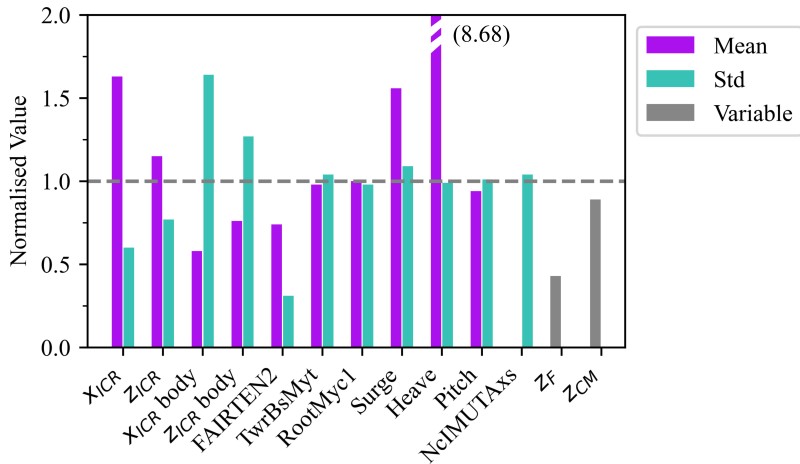

**Figure 25.** The responses of the modified spar FOWT design. Values normalised with the values of the original design as $1 + (x_{adjusted} - x_{original})/|x_{original}|$.

correlated with all responses. Therefore, the fairlead height changes that lead to loads reduction through increasing the $x_{ICR}$ would, at the same time, increase the loads due to decreasing the $z_{ICR}$. Based solely on the correlation coefficients, it is not

possible to determine the relative magnitudes of the two effects and the net effect.

Therefore, an additional step is introduced to select the variables to adjust. Rather than comparing the correlation coefficients, we follow the procedure:

1. For each pair of design variables – ICR statistic, fit a linear function by the ordinary least squares linear regression method. The slope represents the strength of the relationship between the metrics. The coefficient of determination of
the prediction represents how closely the data follows the linear trend.

2. Multiply the slope ($\alpha_{i,j}$) with the coefficient of determination ($R^2$) to obtain a score that represents the strength of the relationship considering the confidence in the coefficient ($x_{i,j}$ in Equation 2).

$$x_{i,j} = \alpha_{i,j} R_{i,j}^2 \tag{2}$$

3. For each pair of the ICR statistic – response statistic, fit a linear function to obtain the slope, coefficient of determination
and their product ($s_{i,j}$ in Equation 3).

$$s_{i,k} = \alpha_{i,k} R_{i,k}^2 \tag{3}$$

4. For each design variable, multiply this product into the corresponding product for the design variables – ICR statistic pair. The resulting score ($f_j$ in Equation 4) represents the potential of the design variable to improve the overall performance

of the design.

$$f_j = \sum_{i=1}^{N_i}\sum_{k=1}^{N_k} x_{i,j} s_{i,k} \tag{4}$$

The subscripts $i$, $j$, and $k$ correspond to the $i$-th ICR statistic ($N_i = 4$), $j$-th design variable, and $k$-th response statistic ($N_k = 8$). The variables/ICR statistics/responses values are normalised by the absolute value of the original design's variable/ICR statistic/response to allow the comparison between different responses while also preserving their signs.

The above procedure yields a list of influence scores allowing the selection of a set of design variables with the most significant overall positive impact on the responses of interest. As reported in Table 16, two design variables have the most impact: the column diameter and mooring line length.

**Table 16.** Influence scores for the semisubmersible's design variables.

| Design variable | Line length | Fairlead Z | Fairlead R | Platform $z_{CM}$ | Column d | Heave plate d |
|---|---|---|---|---|---|---|
| $f_j$ | -1.49 | -0.39 | -0.35 | 0.92 | -1.35 | 0.35 |

Therefore, by consulting Figures 20 and 22, the design task will be to:

– Increase the offset column diameter and mooring line length to increase the mean of both ICR coordinates.

– This should lead to a reduction of all loads.

– Adjusting the variables leads to an undesirable increase in the standard deviation of $z_{ICR}$, as it might increase dynamic mooring tension, and a desirable increase of the mean of $x_{ICR}$, which tends to improve the dynamic tension. The influence scores analysis gives confidence that the net effect should be a lower load.

The design space and the above steps are visualised in Figures 23 and 26.

The comparison of the original and modified designs in Table 17 and Fig. 27 shows a significant reduction in all loads due to the increase in mooring line length and offset column diameter (note that this case study only attempts a "manual" design sizing for simplicity and clear demonstration, and is not an optimisation).

While the correlation coefficients and influence scores were useful in determining the general directions that potentially improve the designs, they gave no insight into the magnitudes of the changes in the ICR and responses. This insight can be gained by examining Figure 26. In particular, it shows that the significant positive correlation between the mooring line length and the mean of $x_{ICR}$ did not significantly contributed to changes in the mean $x_{ICR}$: although the two values change in the same direction, the gradient of ICR by line length is relatively small in comparison to the gradient of ICR by column diameter (see the pattern in the left top subplot). However, the small increase of the line length still led to significant reduction of the dynamic mooring tension through a mechanism different than the ICR position (longer catenary lines provide less stiffness). This highlights an important point: while ICR position can be a useful design

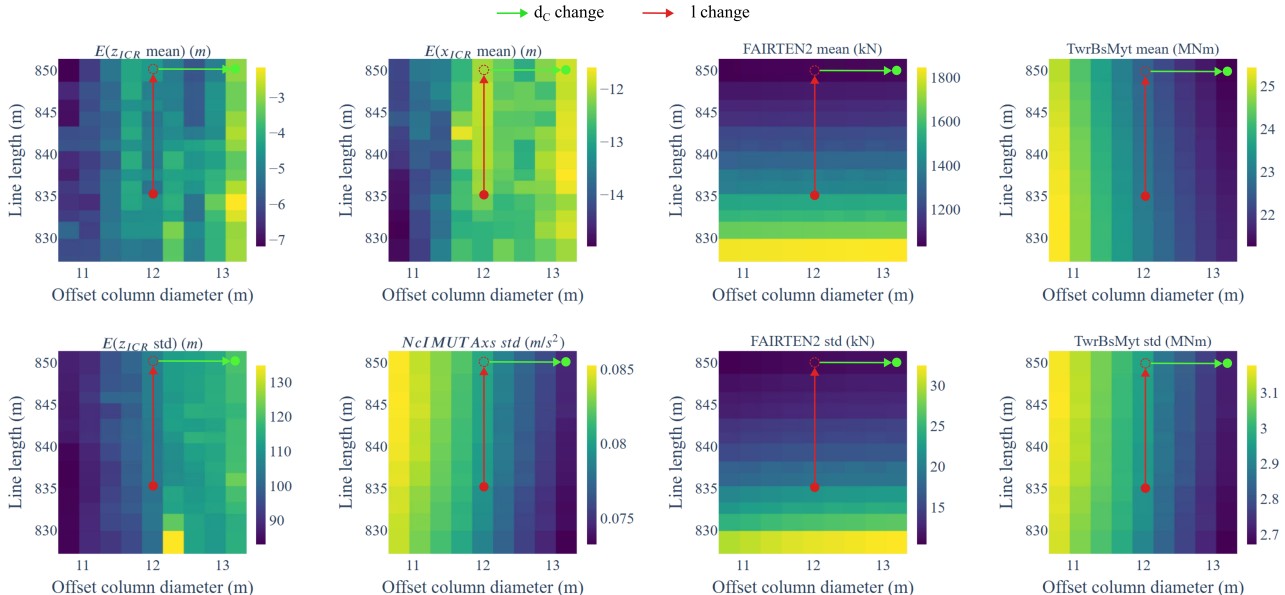

**Figure 26.** Design space visualisation: ICR and response statistics as functions of two selected design variables for the semisubmersible-type FOWT.

**Table 17.** Comparison of the original and the ICR-informed adjusted semisubmersible FOWT designs' responses. Body-fixed frame ICR values in brackets.

| Metric | Unit | Mean | | Std | |
|---|---|---|---|---|---|
| | | Original | Adjusted | Original | Adjusted |
| $x_{ICR}$ | m | -12.13 (-5.85) | -11.86 (1.43) | 48.01 (47.85) | 45.93 (45.79) |
| $z_{ICR}$ | m | -4.76 (-4.98) | -3.06 (-3.12) | 103.28 (103.35) | 119.65 (119.71) |
| FAIRTEN2 | N | 1.45E+06 | 1.05E+06 | 1.98E+04 | 1.12E+04 |
| TwrBsMyt | kN m | 2.30E+04 | 2.12E+04 | 2.93E+03 | 2.71E+03 |
| RootMyc1 | kN m | 3.81E+03 | 3.80E+03 | 2.23E+02 | 2.23E+02 |
| Surge | m | 6.16 | 13.22 | 0.10 | 0.11 |
| Heave | m | -0.01 | 0.09 | 0.03 | 0.02 |
| Pitch | ° | 0.98 | 0.69 | 0.07 | 0.06 |
| NcIMUTAxs | $m s^{-2}$ | - | - | 0.08 | 0.07 |

metric, it is only one of many metrics. In this case, line length was a parameter which allowed the reduction of mooring tension without negatively affecting the ICR position.

This statement is also supported by an examination of a similar plot, but this time based on variations of the parameters that do not directly affect the mooring design or loads. Figure 28 presents the column diameter and heave plate diameter design space. In the absence of mooring parameters changes, the effect of varying the ICR position in relation to the

**Table 18.** The original and the ICR-informed adjusted semisubmersible FOWT designs features.

| Feature | Unit | Original | Adjusted |
|---|---|---|---|
| Offset column diameter | m | 12.0 | 13.20 |
| Line length | m | 835.5 | 850.0 |

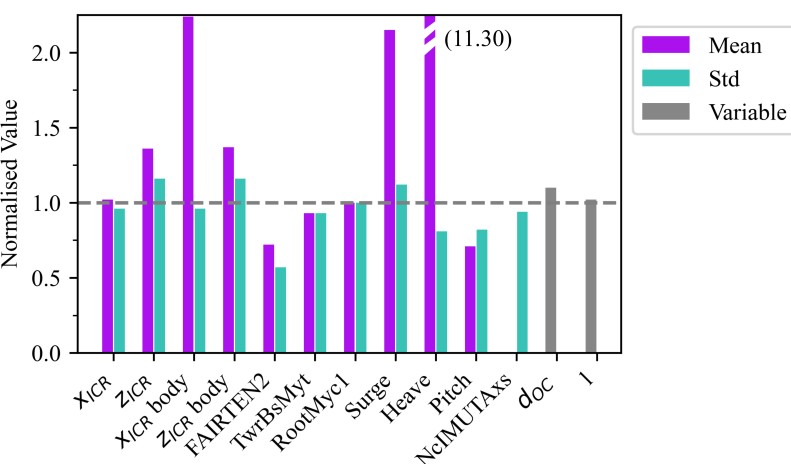

**Figure 27.** The responses of the modified semisubmersible FOWT design. Values normalised with the values of the original design as $1 + (x_{adjusted} - x_{original})/|x_{original}|$.

tower top and fairlead positions is more pronounced. Namely, as the distance between the $z_{ICR}$ and fairlead position ($z = -14$ m) decreases, the dynamic tension decreases. At the same time, the distance between the $z_{ICR}$ and the tower top increases, increasing the tower top acceleration and so the bending moment experienced at the base of the tower.

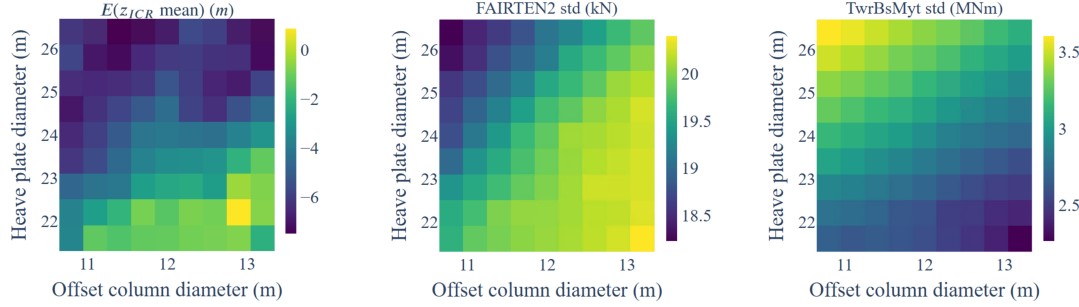

**Figure 28.** Additional design space visualisation: ICR and response statistics as functions of the offset column and heave plate diameter of the semisubmersible-type FOWT.

# 7 Conclusions

This study investigated the instantaneous centre of rotation (ICR) of spar and semisubmersible floating offshore wind turbines (FOWTs) through time-domain analysis. It examined the dependence of ICR on complex environmental loading conditions and design features of the support structure, as well as the relationships between the ICR statistics and key FOWT responses.

The statistics of ICR coordinates under increasingly complex external loading conditions were analysed, isolating the impacts of mean wind speed, shear, turbulence intensity, regular wave frequency, irregular waves, and current speed. The results indicated that the ICR coordinates of FOWTs operating in realistic environmental conditions exhibit significant variability, following a heavy-tailed, sharp-peaked distribution with a high likelihood of extreme values. Notably, at a moderate confidence level of $60\%$, the range of the vertical ICR coordinate of the spar-type FOWT was approximately $80\,\mathrm{m}$, indicating that precise placement of the ICR is challenging in complex environments.

To inform practical design applications, we explored which design variables could be adjusted to influence ICR statistics and how these statistics affected key responses, for each substructure type separately. We identified the design variables most correlated with ICR: the platform's center of mass and the vertical position of the fairlead for the spar, as well as the mooring line length, offset column diameter, and heave plate diameter for semisubmersible FOWTs. A full factorial design of experiments revealed significant correlations between ICR and the responses of interest: nacelle acceleration, tower base fore-aft bending moment, blade root out-of-plane moment, and mooring line tension. For the spar system, these were most significantly correlated with the horizontal ICR coordinate, and for the semisubmersible, the highest correlation was found with the mean of the z-coordinate of the ICR.

Sets of highly correlated design variables, ICR statistics, and responses were selected and used to formulate case studies, where 2D design trends were visualised and analysed to identify preferred design directions. In the spar FOWT case, the fairlead position was adjusted to shift the mean of $x_{ICR}$ closer to the fairleads. Although it was impossible to align the mean ICR with the fairleads with the design modifications considered, moving the ICR in that direction significantly reduced mean and dynamic mooring loads. This adjustment had a negative impact on the tower and blades loads, which were mitigated by shifting the platform's centre of mass, thereby shifting the standard deviation of $z_{ICR}$ and restoring these loads to their original levels. Even though the ICR is dynamic and exhibits significant variability, the case study showed that shifting the mean ICR towards specific locations can significantly reduce structural loads.

The semisubmersible case study demonstrated that analysing the correlations between the variables, ICR, and responses is not always sufficient to define a good design direction. To address more complex dependencies, we introduced an additional metric: the influence factor, which, based on the slope and coefficient of determination of a linear fit, quantifies the overall impact of a design variable on all responses of interest. By adjusting the two variables with the highest influence factors, we were able to achieve a design in which all responses of interest were improved.

The results demonstrated the effect of the ICR on the trade-off between dynamic loads at different locations on the structure. Shifting the ICR closer to the tower top reduced nacelle acceleration and tower base loads (which are driven by acceleration). In contrast, shifting the ICR closer to the fairleads reduced mooring line loads. Importantly, the study showed that the ICR should not be treated as an independent design variable. It can only be adjusted by modifying other design variables that also impact the loads through mechanisms different from the ICR itself, and in some cases, may even counteract its effects. Because of these complexities, relying solely on the ICR to guide the design is insufficient. Instead, the ICR is useful for understanding one of the mechanisms that affect the global response and loads experienced by FOWTs.

## 8  Future work

To generalise the conclusions to any floating system, further research should investigate how the characteristics of externally applied loads – such as amplitude, point of application, distribution, and frequency – along with the system's inertial and restoring properties affect the motion response, ICR, and structural loads. This research should abstract from platform geometry and environmental conditions, focusing instead on the fundamental load-response interactions. Additionally, the parametrisation of the FOWT support structure should be refined to allow more control over the designs and the ICR, in particular, the separation of the semisubmersible columns, which is an important design feature overlooked in this study. Conducting the analysis in the design load cases commonly considered at the concept design/feasibility study stage would be insightful. Lastly, a more efficient, frequency-domain approach to computing the ICR would benefit initial design space screening and optimisation tasks.

### Appendix A

This appendix presents the Spearman correlation coefficient calculation for an example response (standard deviation of the tower base bending moment) and an example ICR metric (standard deviation of the $z$-coordinate) using Equation 1 on a subset of data analysed in this paper. The differences of the ranks ($d_i$ in Table A1) are summed to obtain the correlation coefficient for the response-ICR metric pair.

### Appendix B

This appendix presents the figures supporting the discussion of Section 3. Figure B1 shows the ICR coordinates distributions for two additional wave periods: $31\,\mathrm{s}$ and $125\,\mathrm{s}$ (pitch and surge natural frequencies). Figure B2 shows the distributions of the horizontal velocity in points A and B for all wave periods.

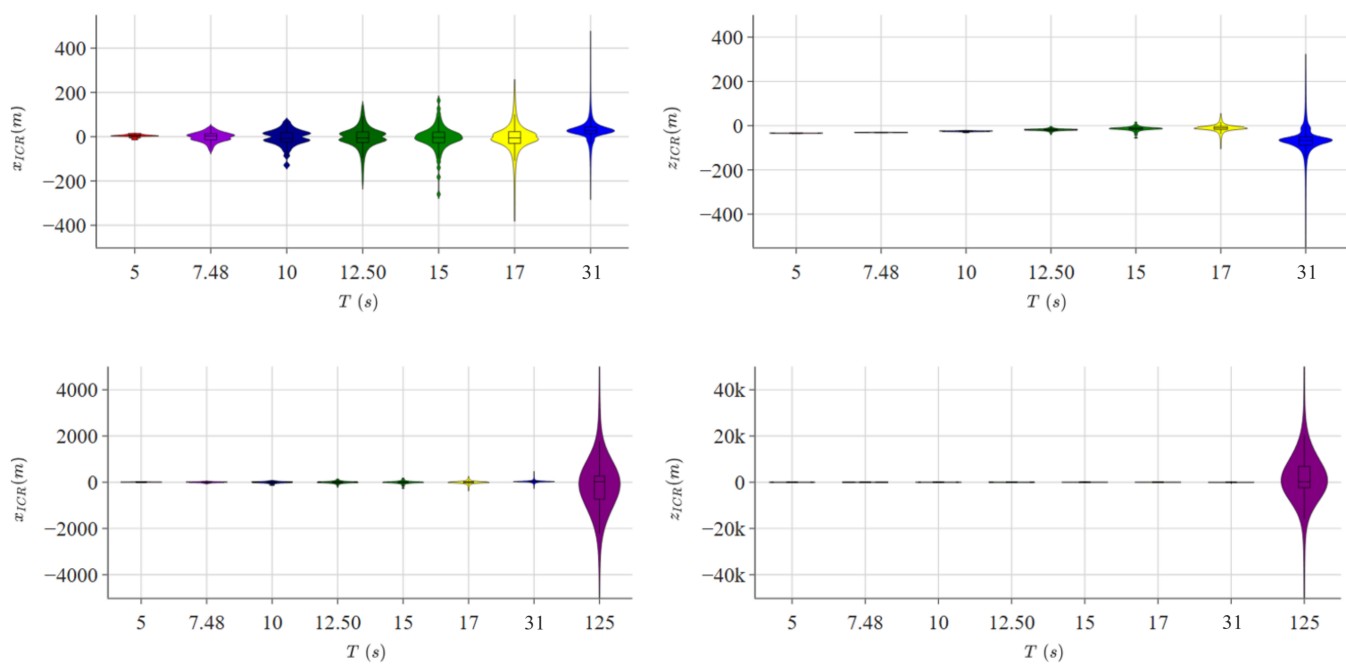

**Figure B1.** Distribution of ICR coordinates in regular waves of periods 5 s–31 s (top) and 5 s–125 s (bottom).

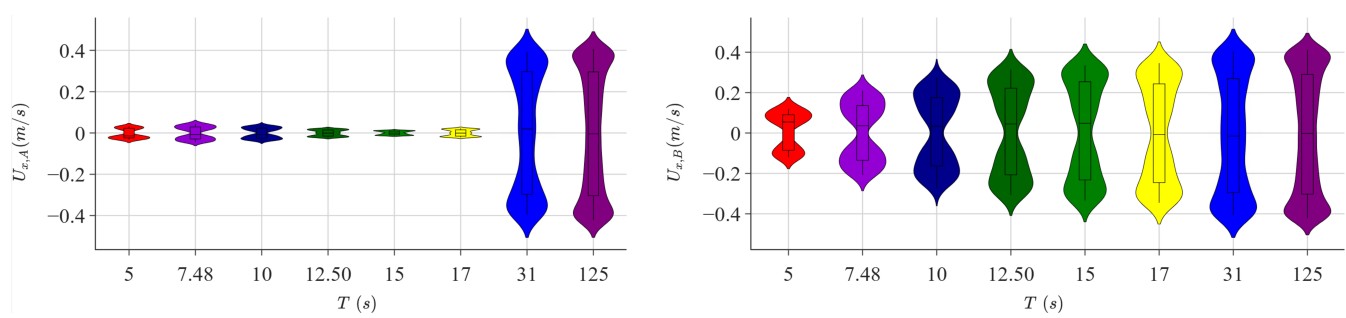

**Figure B2.** Distributions of the horizontal velocity in points A (left) and B (right).

**Table A1.** Spearman rank correlation calculation example.

| Sample | My std (kN m) | Rank | $z_{ICR}$ std (m) | Rank | $d_i$ |
|--------|---------------|------|-------------------|------|-------|
| 1 | 2.65E+03 | 5 | 120.55 | 4 | 1 |
| 2 | 2.66E+03 | 4 | 119.29 | 5 | -1 |
| 3 | 2.87E+03 | 3 | 126.24 | 2 | 1 |
| 4 | 3.21E+03 | 2 | 122.14 | 3 | -1 |
| 5 | 3.72E+03 | 1 | 149.58 | 1 | 0 |

## Appendix C

This appendix presents the correlation matrices (Spearman correlation) between the design variables and the responses of interest for the spar (Fig. C1) and semisubmersible (Fig. C2) floating offshore wind turbines. For clarity, the correlation scores below the absolute value of 0.4 are not shown. Notice the primary impact of the mooring design on the mean response and the floating platform design on the dynamic response.

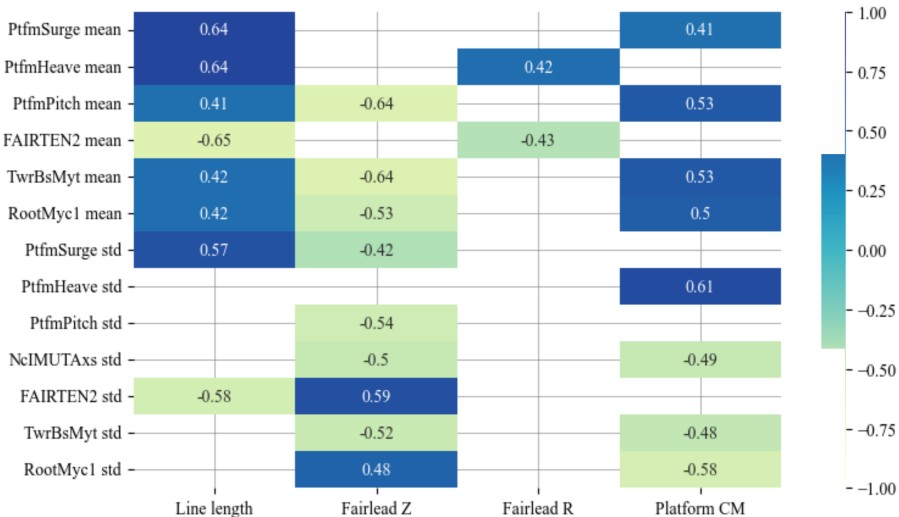

**Figure C1.** Correlation matrix between the design variables and the responses of interest for the spar design.

*Author contributions.* KP, MC, JJ, GB, MH, DZ conceptualised the study. KP developed the methodology and performed formal analysis, investigation, visualisation and original draft preparation. MC, JJ, GB, MH, DZ, AC provided supervision. All authors contributed to the draft review and editing.

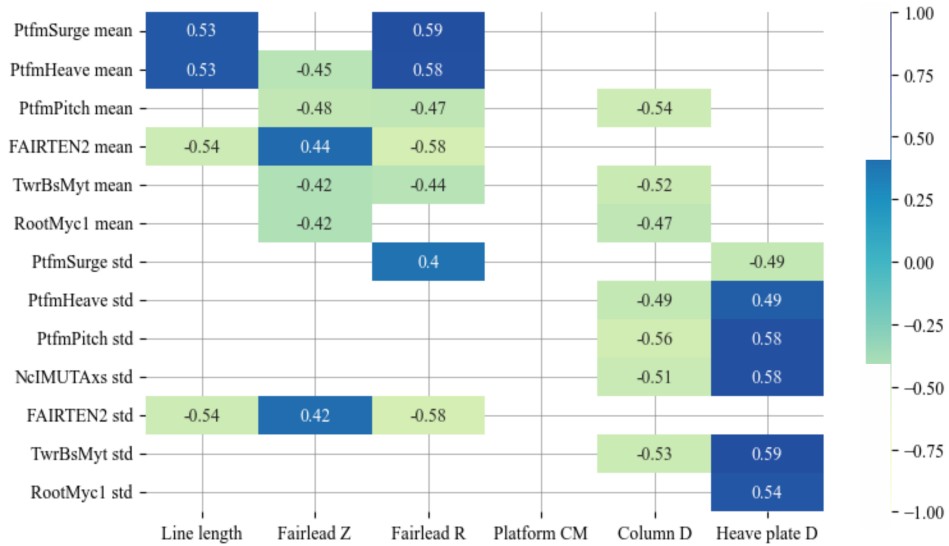

**Figure C2.** Correlation matrix between the design variables and the responses of interest for the semisubmersible design.

*Competing interests.* Some authors are members of the editorial board of this journal.

*Acknowledgements.* This work was supported by the University of Strathclyde REA 2024.

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
