# Peer review of "Investigation into instantaneous centre of rotation for enhanced design of floating offshore wind turbines"

_Wind Energy Science, 2024_

## Author Comment (AC2)

**Response to the Reviewer's comments - review 1**

Investigation into Instantaneous Centre of Rotation for Enhanced Design of Floating Offshore Wind Turbines.

Dear Reviewer,

The authors would like to express their gratitude to the Reviewer for the thorough review and constructive comments. Please find our response with an indication and details of the modifications made below.

**Comment 1** "(...) it does not seem appropriate to 'optimise' a design using this parameter. It doesn't seem possible to clearly identify the most advantageous (=optimal) ICR position, as it is very dependent on the floater concept and the response parameters you want to optimise."

**Response 1** Thank you for your comment. You are correct that the ICR is not a parameter that can be directly used to optimise the floating offshore wind turbine (FOWT) design in a strict sense. The optimal position of the ICR does not directly correlate to the "best" design. However, it serves as a useful metric, and adjusting its location (or, statistics of it) can lead to significant improvements in multiple response parameters. Shifting the ICR towards a particular location may not necessarily minimise loads, but it will help reduce them. The ideal location of the ICR (or its statistics) is indeed concept-specific, as demonstrated in the paper. This varies between different substructure concepts, such as spar and semi-submersible, due to differences in the relative importance of external loads and stability mechanisms. The methodology is intended to assist in the early-stage design of FOWTs, providing additional insights into system behaviour and is meant to be applied only within the context of a specific floating substructure concept. It should not be used to choose between different concepts, but rather to adjust design parameters (e.g., column diameter, mooring line length) within a single concept.

**Modification 1** A statement clarifying the use of ICR in design context was added to the Conclusion section: "Importantly, the study showed that the ICR should not be treated as an independent design variable. It can only be adjusted by modifying other design variables that also impact the loads, through mechanisms different from the ICR itself, and in some cases, may even counteract its effects. Because of these complexities, relying solely on the ICR to guide the design is insufficient. Instead, the ICR is useful for understanding one of the mechanisms that affect the global response and loads experienced by FOWTs".

**Comment 2** "The correlations between the design parameters, the centre of rotation and the response parameters are not very clear, they seem to depend strongly on the floater concept and configuration. This means that the correlations presented in the paper probably cannot be generalised or applied to another floater design. They would have to be recalculated for each design, which is likely to be a time-consuming task. The practical application of this aspect of the study may therefore be limited. I suggest to acknowledge this limitations of the study related to practical application explicitly in the Conclusions section of the manuscript. This would help to prevent any "unwise" application by less experienced engineers. See also specific comment below".

**Response 2** That is correct, the ICR trends and correlations with the design features and responses of interest strongly depend on the floating system concept and configuration and cannot be generalised to all FOWT types. This methodology is intended to aid improvements of a single-class FOWT, and not to compare the designs of various classes.

**Modification 2** While the Conclusions section already summarises the findings for each floating concept separately, we have added a statement to clarify this point further: "To inform practical design applications, we explored which design variables could be adjusted to influence ICR statistics and how these statistics affected key responses, **for each substructure type separately**". Additionally, we changed the last sentence of the Introduction section to reflect that the methodology and not the specific findings is the main added value of this work: "The **methods proposed by this paper** can be applied in a practical design scenario, as will be demonstrated in Sect. 6".

**Comment 3** "(...) plotting the ICR over different wave periods could be a useful figure, together with the motion RAOs and other hydrodynamic parameters".

**Response 3** Thank you for this suggestion. The study of the ICR versus wave frequency and amplitude was conducted in a previously published work:

> *Patryniak, K., Collu, M., and Coraddu, A.: Rigid body dynamic response of a floating offshore wind turbine to waves: Identification of the instantaneous centre of rotation through analytical and numerical analyses, Renewable Energy, 218, 2023.*

We acknowledge that studying the ICR in relation to motion response RAOs and load parameters would be very interesting. We consider this to be a significant extension of the current study, and we have recommended it for further investigation in the Future Work section.

**Comment 4** "Line 151-154: Change of underwater geometry is (together with the mooring system design parameters) the main design variable for a FOWT (not considering the WTG design). So, excluding the change of hydrodynamic loading on the structure seems to be a serious limitation".

**Response 4** Following the feedback received, we studied in more detail the impact of the hydrostatic stiffness and potential hydrodynamic coefficients on the ICR and responses of the semisubmersible designs range considered in this study. This analysis showed a significant impact, as indicated by the reviewers. Therefore, we updated the numerical model to update the hydrostatic stiffness matrix and potential coefficients for each semisubmersible platform design based on the Boundary Element Method solution. Additionally, we introduced re-ballasting to keep the draft of different platforms the same. Sections 4–6 with now present the updated results. Thank you for highlighting this important point.

**Modification 4** Changes were made to numerical results in figures, tables and text of Sections 4–6. In particular, the new results changed the shape of the second case study and its conclusions, which necessitated changes in the Conclusion section. In particular:

- Identifying a single ICR metric to shift to improve all responses of interest was no longer possible,

- Identifying a clear design direction was not possible, therefore we proposed the use of "influence factors", as explained in the updated draft.

- We summarised the limitations of the methodology that became more apparent after the updated analysis.

Please refer to the updated draft for all modifications.

**Comment 5** "Line 166-167: FOWT are designed to a large extend for ultimate limit strength (ULS) and fatigue limit strength (FLS). The proposed averaging process seems to be targeted at "representative" FLS responses. However, ULS responses (maximum responses within all conditions) would be averaged by this process and thus largely underestimated".

**Response 5**    In this work, the authors decided to average the results out between a set of FLS conditions, with some ULS-like conditions (maximum thrust conditions around rated wind speed) contributing to the average significantly (please refer to Table 1). However, the methodology presented in the paper can be adapted to account for another set of loading conditions, depending on the design scenario. As noticed, the results and conclusions are only valid within these conditions.

**Comment 6**    "Line 167-168: Are the same seeds used for each design variant? If not results between design variants would not be comparable with each other".

**Response 6**    Yes, all results were obtained with the same random seed.

**Modification 6**    We added a clear statement of the above in Section 2.3 Dynamic simulations: "Each simulation is run for 1 hour (excluding the initial transient phase) with a single random seed **(the same seed in all simulations)**.

**Comment 7**    "Figure 17: Why do the original design values show different ICR values? Shouldn't all original design values share the same ICR value? The original design values should all represent the same original design configuration and thus have same result values".

**Response 7**    Thank you for noticing this, that is correct. The results were generated by systematic variations of the variables between the minimum and maximum values and the original design values were not always part of the design matrix; the markers corresponding to the original designs were mislocated.

**Modification 7**    We have now added the results of the original design and marked the correct original design values in Figures 19 and 20 (note that these are also updated in response to other comments).

[Figure]

Figure 1: ICR statistics for different one-at-a-time design variable perturbations for a spar FOWT. Original design values marked with black outline.

**Comment 8**    "Table 10. (FAIRTEN2 mean Value 10983.62): This seems to be a very high mean/static tension especially, considering that usually the dynamic tension is the governing mooring line maximum load."

[Figure]

Figure 2: ICR statistics for different one-at-a-time design variable perturbations for a semisubmersible FOWT. Original design values marked with black outline.

**Response 8**   The high value of mean tension results from an unusual combination of a very low fairlead radius and short mooring lines, leading to very tight lines. Even though this configuration is beyond the usual design choices, it is included in the full factorial study, as each of the fairlead radius and line length values could be good values in combination with other, less extreme design parameters.

The semisubmersible results were re-computed as per Comment 4 of Reviewer 1 and Comment 5 of Reviewer 2. The updated Section 5 now uses a design matrix where the fairlead height is not varied and the tension values are less extreme.

**Modification 8**   No change has been made to the numerical data of Table 10. However, the order of columns in the table was changed, as it was noticed that the original table presented results related to each of the concepts under the wrong headings ("Spar", "Semisubmersible"). Additionally, the values were updated as per other comments.

**Comment 9**   "Table 11. (Platform zCM adjusted value -100.00): The vertical CM of a SPAR is usually already as low as technically possible. Thus, reducing the CM without any major changes to geometry seems unlikely. Then, if geometry would change, hydrodynamic loading would change and the correlations might be different".

**Response 9**   Thank you for this practical observation. We believe that the OC3 Hywind spar is likely not an optimised design, being one of the first FOWT designs out there, therefore there should be some room for higher draft design variations, subject to port facilities and other practical constraints. However, we do agree that additional changes would be necessary to accommodate the change of the centre of gravity. Since a spar is primarily ballast-stabilised, we believe that increasing the draft of the structure would not significantly affect the motion response and structural loads, the shift of the centre of mass being the dominant effect.

**Modification 9**   We added a note clarifying this point in section 6.1 Spar sizing design case study: "Implementing such adjustments may necessitate additional design modifications—for example, lowering the centre of mass could require a deeper draft structure. While some consideration has been

given to the practicality of these design variations, their overall feasibility is not guaranteed".

**Comment 10** "Table 13. (Offset column d, mean Re-vol. value 8.38): Offset column has a major impact on hydrostatic stiffness and restoring. Reducing the parameter by this extend is would probably result in insufficient floating stability thus large tilt during power production and even risk of capsizing/downflooding".

**Response 10** We double-checked the maximum pitch angle from the simulations of the different variants of the semisubmersible, as presented in the figure below. All designs had sufficient stability in all conditions considered (including that of maximum thrust).

[Figure]

**Once again, we would like to thank the reviewer for the thorough review of the article and extremely useful feedback. We believe that this helped to make the paper better, and we greatly appreciate your time spent reviewing it and the useful input you shared.**

---

## Author Comment (AC3)

**Response to the Reviewer's comments - review 2**

Investigation into Instantaneous Centre of Rotation for Enhanced Design of Floating Offshore
Wind Turbines.

Dear Reviewer,

The authors would like to express their gratitude to the Reviewer for the thorough review and
constructive comments. Please find our response with an indication and details of the modifications
made below.

**Comment 1**   "(...) the paper fails in presenting very important data and equations needed for a
proper interpretation of the results and conclusions".

**Response 1**   Thank you for the thorough review of the draft with respect to the interpretability
of the results and conclusions. We made appropriate modifications to improve on this point, as
detailed in the remainder of this document.

**Comment 2**   "(...) some problematic theoretical assumptions are adopted which, in this reviewer's
opinion, can significantly impair the credibility and reproducibility of the results".

**Response 2**   Based on the reviewers' comments, we thoroughly analysed the impact of the as-
sumptions on the results. As a result, some of the assumptions were relaxed and parts of the analysis
were revised, as detailed in the remainder of this document.

**Comment 3**   "Since the analyses and conclusions are highly dependent on the dynamics of both
floaters, the authors must introduce a section with detailed data for each of the FOWT models used
in the paper. The data has to include at least: mass and moments of inertia (or radii of gyration),
centre of gravity, centre of buoyancy, metacentric height, restoring matrix, added mass, Morison
model parameters (element length/diameters and drag coefficients), and natural periods for the
relevant degrees of freedom. Also, some data about the wind turbine (thrust at rated wind speed,
RNA mass), and mooring system (chain properties, pretension, anchor radius), are also necessary.
Without this information, it is not possible for the reader to evaluate if the results presented later
in the paper are compatible with the conclusions. For example, how can the reader understand the
impact of heave plate size in the ICR coordinates response, if there is no explanation about how
the heave plates are modelled? In addition, the results are not reproducible. Please note that it is
not enough to just refer to the original publication describing the floater. This information shall be
shown in the present article".

**Response 3**   Thank you for this suggestion. To support the analysis, we provided the additional
information as requested.

**Modification 3**   We added Subsection 2.5 Floating platform and turbine characteristics with two
new tables: one with the comparison of the important system characteristics of each reference design,
and one with the properties of the turbine.
"The analysis is performed for two FOWT concepts: the OC3 Hywind spar (Jonkman, 2010)
and OC4 semisubmersible (Robertson et al., 2014), both with the 5 MW reference rotor (Jonkman
et al., 2009). The details of the reference designs are given in Tables 2–3 and Figure 7."
Additionally, we detailed the Morison model parameters in Section 2.3 Dynamic simulations.

**Table 2.** OC3 Hywind Spar and OC4 semisubmersible FOWT original designs characteristics. CG – centre of gravity; CB – centre of buoyancy.

| Property | Unit | OC3 spar | OC4 semisubmersible |
|---|---|---|---|
| **Platform, including ballast**: | | | |
| Draft | m | 120 | 20 |
| Diameter | m | 6.5–9.7 (tapered) | - |
| Offset column diameter | m | - | 12 |
| Base column diameter | m | - | 24 |
| Mass | kg | $7.47 \cdot 10^6$ | $1.35 \cdot 10^7$ |
| Vertical CG | m | $-89.92$ | $-13.46$ |
| Pitch inertia about CG | $kg\,m^2$ | $4.23 \cdot 10^9$ | $6.83 \cdot 10^9$ |
| **Hydrostatics**: | | | |
| Vertical CB | m | $-61.17$ | $-13.17$ |
| Metacentric height | m | 28.7 | 10.7 |
| Stiffness coefficient: | | | |
|   Heave | $N\,m^{-1}$ | $3.33 \cdot 10^5$ | $3.84 \cdot 10^6$ |
|   Pitch | $N\,m\,rad^{-1}$ | $-4.99 \cdot 10^9$ | $-3.78 \cdot 10^8$ |
| **Infinite-frequency added mass**: | | | |
| Surge-surge | kg | $7.569 \cdot 10^6$ | $6.49 \cdot 10^6$ |
| Heave-heave | kg | $2.35 \cdot 10^5$ | $14.70 \cdot 10^6$ |
| Pitch-pitch | $kg\,m^2$ | $3.7 \cdot 10^{10}$ | $7.21 \cdot 10^9$ |
| Surge-pitch | $kg\,m$ | $-4.71 \cdot 10^8$ | $-8.51 \cdot 10^7$ |
| **Mooring system**: | | | |
| Fairlead height | m | $-70$ | $-14$ |
| Fairlead radius | m | 4.7 | 40.87 |
| Water depth (anchor position) | m | $-320$ | $-200$ |
| Anchor radius | m | 853.87 | 837.6 |
| Unstretched line length | m | 902.20 | 835.5 |
| Chain diameter (volume-equivalent) | m | 0.09 | 0.077 |
| Pretension | kN | 910 | 1100 |
| **Natural frequency**: | | | |
| Surge | Hz | 0.008 | 0.009 |
| Heave | Hz | 0.032 | 0.058 |
| Pitch | Hz | 0.034 | 0.037 |

**Table 3.** NREL 5 MW reference rotor characteristics.

| Property | Unit | Value |
|---|---|---|
| RNA mass | kg | 350000 |
| Hub height | m | 90.0 |
| Cut-in, rated, cut-out wind speed | $m\,s^{-1}$ | 3, 11.4, 25 |
| Rated wind speed thrust | kN | 805.11 |

"The first-order potential hydrodynamics of the spar platform are modelled through frequency-to-time-domain transforms based on the potential coefficients obtained from the boundary element method (BEM) code WAMIT (Lee and Newman, 2006). Viscous loads are computed from Morison's theory. The semisubmersible platform's offset columns, heave plates, and main column are modelled using a hybrid approach (potential and Morison). The potential coefficients are obtained with pyHAMS BEM solver (NREL, 2025), as implemented in RAFT (Hall et al., 2022). The slender pontoons and cross-braces (diameter: 1.6 m) are treated with a Morison-only approach, with hydrodynamic coefficients listed in Table 1. Second-order hydrodynamics are not considered."

**Table 1.** Hydrodynamic model parameters.

| Parameter | Unit | OC3 spar | OC4 semisubmersible |
|---|---|---|---|
| Maximum Morison element size | m | 0.5 | 1 |
| Maximum BEM panel size | m | 1 | 1.5 |
| Added mass coefficient | - | - | Pontoons and cross-braces: 0.63 |
| Transverse viscous-drag coefficient | - | 0.6 | Main column: 0.56 |
| | | | Offset column: 0.61 |
| | | | Heave plate: 0.68 |
| | | | Pontoons and cross-braces: 0.63 |

**Comment 4** "In addition, it is necessary to include the equation of motions, indicating how the variation of the design variables affect the dynamics of the floaters. For example, how does the variation in the heave plates affect the floater response? Which forces are affected by variations in the column diameters? It is impossible understand the impact of the design variations in the responses without knowing how the dynamics are modelled".

**Response 4**

Our understanding is that the reviewer requests the analytical equations of the dynamic model to be presented directly in the paper, together with the analysis of the impact of the design variables on these equations.

The analysis relies on a comprehensive time domain model OpenFAST where the equations of motion are nonlinear and expressed in a modular generalised state-space form with module-level states (including constraints), inputs, and outputs, with module-to-module coupling handled through input-output constraints, and where the various state-space equations depend on the set of parameters (including design variables). Therefore, providing the exact equations for loads and motion response is not feasible. For example, the equation for inertia force at a heave plate node alone:

$$
\vec{F}_I^s = \left\{ \begin{array}{c} 0 \\ 0 \\ C^s \left\{ \begin{array}{c} C_{P_{dx}} \pi \left(R^s + t_{MG}^s\right)^2 \\ p_{dnn} \end{array} \right\} + \dfrac{C_{A_{dx}} \rho_W \overrightarrow{V_n} \left(\vec{a}_f \bullet \overrightarrow{V_n}\right)}{\sqrt{\left(\overrightarrow{V_n} \bullet \overrightarrow{V_n}\right)}} \\ 0 \\ 0 \\ 0 \end{array} \right\}
\tag{1}
$$

Given the complexity, since this paper does not focus on the development of the model, we are inclined not to include the exact equations. Instead, we propose a range of modifications outlined below.

Having said that, we agree that it would be of added value to present a systematic study of the variables-load-response effects, and to show how the design variations affect the load distribution,

and how this affects the motion response and, in turn, the ICR. However, such systematic analysis would be a substantial study on its own and did not fit within the project timelines. Therefore, we recommended it as future work, as mentioned in the Future work section:

"To generalise the conclusions to any floating system, further research should investigate how the characteristics of externally applied loads – such as amplitude, point of application, distribution, and frequency – along with the system's inertial and restoring properties affect the motion response, ICR, and structural loads. This research should abstract from platform geometry and environmental conditions, focusing instead on the fundamental load-response interactions".

**Modification 4**

- To maintain the focus of the paper on ICR analysis within the design context and avoid duplicating information already published elsewhere, we reference the original papers to present the equations behind OpenFAST and the impact of the design variables on the dynamic response:

  - *Jonkman, J. M. "The New Modularization Framework for the FAST Wind Turbine CAE Tool." 51st AIAA Aerospace Sciences Meeting including the New Horizons Forum and Aerospace Exposition, 7–10 January 2013, Grapevine (Dallas/Ft. Worth Region), TX [online proceedings]. URL: http://arc.aiaa.org/doi/pdf/10.2514/6.2013-202. AIAA-2013-0202. Reston, VA: American Institute of Aeronautics and Astronautics, January 2013; NREL/CP-5000-57228. Golden, CO: National Renewable Energy Laboratory.*

  - *Jonkman, J. M.; Branlard, E. S. P.; and Jasa, J. P. "Influence of Wind Turbine Design Parameters on Linearized Physics-Based Models in OpenFAST." Wind Energy Science. Vol. 7, No. 2, March 2022, pp. 559-571; NREL/JA-5000-81481. Golden, CO: National Renewable Energy Laboratory; DOI: 10.5194/wes-7-559-2022.*

- Throughout the paper, we emphasize how changes in specific design and load characteristics influence motion and ICR behaviour.

- The hydrodynamic model description in Chapter 2.3 was improved to better outline how each member is modelled.

- In Appendix C, we provide correlation matrices based on our results that show the links between the design variables, loads and responses empirically.

**Comment 5** "The paragraph starting at line 151 says the potential-theory hydrodynamic coefficients are not re-calculated for the design variations. The justification given for this assumption is that the variations of underwater floater dimensions are kept "small". This reviewer believes that this is a very problematic assumption. The variations in diameter for both the spar (at the waterline) and semi (columns) will significantly affect both the added mass and the excitation force, which are directly related with the results and conclusions. Even more serious may be the variation in the heave plates' diameter (from 19.33 m to 27.35 m), which will very significantly affect the heave and pitch added masses, impacting the floater dynamics substantially. Doing a very simple approximation for the heave added mass of each column + heave plate set, as the mass of a semi-sphere with the heave plate diameter ($A33 = pi * rho * D^3/12$): we have 1.9e6 kg for D = 19.33 m, and 5.5e6 kg for D = 27.35. Is it reasonable to neglect the added mass variations in this case? If the authors are not able to run the potential theory analysis for each design variation, an alternative could for example be to use scaling rules for estimating the radiation and diffraction loads of the design variations, and then re-run the analyses with the scaled loads. If the authors prefer not to re-run analyses, they at the very least must be able to discuss how this assumption (keeping the

potential theory loads unchanged for all design variations) will affect the results – and why they think the conclusions are still valid, after adopting the assumption".

**Response 5**   Following the feedback received, we studied in more detail the impact of the hydrostatic stiffness and potential hydrodynamic coefficients on the ICR and responses of the semisubmersible designs range considered in this study. This analysis showed a significant impact, as indicated by the reviewers. Therefore, we updated the numerical model to update the hydrostatic stiffness matrix and potential coefficients for each semisubmersible platform design based on the Boundary Element Method solution. Additionally, we introduced re-ballasting to keep the draft of different platforms the same. Sections 4–6 with now present the updated results. Thank you for highlighting this important point.

**Modification 5**   Changes were made to numerical results in figures, tables and text of Sections 4–6. In particular, the new results changed the shape of the second case study and its conclusions, which necessitated changes in the Conclusion section. In particular:

- Identifying a single ICR metric to shift to improve all responses of interest was no longer possible,

- Identifying a clear design direction was not possible, therefore we proposed the use of "influence factors", as explained in the updated draft.

- We summarised the limitations of the methodology that became more apparent after the updated analysis.

Please refer to the updated draft for all modifications.

**Comment 6**   "Line 71: You say that the method proposed in the paper is "readily applicable" in a practical design. That may be a bit too optimistic, since the analysis may be quite cumbersome. Please re-word".

**Modification 6**   The end of Section 1 was modified as follows: "The methods proposed by this paper can be applied in a practical design scenario, as will be demonstrated in Sect. 6".

**Comment 7**   "Line 107: Please explain what is the meaning of "direct proportion"".

**Response 7**   We added a definition of the "direct proportion" term in the footnote.

**Modification 7**   Footnote text added: "Direct proportion is a relation between two quantities where the ratio of the two is equal to a constant value".

**Comment 8**   "Lines 117/118: Seems like the references are flipped (regular waves should be Fig 5, irregular waves should be Fig 6)".

**Response 8**   Figures 3 and 4 present the time history and distribution of ICR in regular waves, while Figures 5 and 6 present the time history and distribution of ICR in irregular waves. The references in lines 117-118 seem to be correct.

**Comment 9**   "Line 124: Please add a reference to the Kolmogorov-Smirnov method".

**Modification 9**   A reference was added.

**Comment 10**   "Line 134: The explanation on how to use the Spearman correlation coefficients is a bit confusing. Can you provide an example, showing how you ranked design variables, ICR coordinates, and FOWT responses?"

**Modification 10**  We added a more detailed explanation of the Spearman correlation coefficient, including references to the design variables, ICR coordinates, and FOWT responses to make it less abstract. Additionally, we included a short example calculation in Appendix A.

Section 2.2 Instantaneous centre of rotation statistics: "To analyse the relationships between design variables, the ICR, and FOWT responses, we compute the Spearman rank correlation coefficient[1], which quantifies the statistical dependence between two variables. As illustrated in Table A1 in Appendix A, for each pair of variables (for example, the standard deviation of the tower base bending moment and the standard deviation of $z_{ICR}$), we list their values from different simulations, then rank each list from smallest to largest (the smallest value in a list gets rank 1, the second smallest value gets rank 2, etc.). The Spearman correlation coefficient $\rho$ is computed based on the sum of the differences between the ranks of the paired values:

$$\rho = 1 - \frac{6 \sum_{i=1}^{n} d_i^2}{n(n^2 - 1)} \tag{2}$$

$d_i$ is the difference between the ranks for sample (simulation) $i$, and $n$ is the number of observations (simulations). Spearman's correlation assesses the strength of a monotonic relationship, making no assumptions about the underlying data distribution. A correlation close to $+1$ or $-1$ indicates a strong monotonic relationship:

- $\rho \approx 1$: as one variable increases, the other tends to increase.

- $\rho \approx -1$: as one variable increases, the other tends to decrease.

- $\rho \approx 0$: little to no monotonic relationship between the variables.

Since Spearman's correlation is based on ranks rather than raw values, it is robust to nonlinearity and less sensitive to outliers than Pearson's correlation. However, it does not imply causation, meaning that a strong correlation does not indicate that changes in one variable directly cause changes in another. An example calculation is provided in Appendix A."

Appendix A: "This appendix presents the Spearman correlation coefficient calculation for an example response (standard deviation of the tower base bending moment) and an example ICR metric (standard deviation of the $z$-coordinate) using Equation 1 on a subset of data analysed in this paper. The differences of the ranks ($d_i$) are summed to obtain the correlation coefficient for the response-ICR metric pair".

**Comment 11**  "Line 143: The rotor speed actually varies due to the fluctuations on the relative incident wind, caused by a combination of wind speed variations and RNA motions".

**Modification 11**  We changed the text in Section 2.3: "The NREL 5 MW baseline variable-pitch variable-speed controller allows for blade pitch and rotor speed variations depending on the mean wind speed and slight rotor speed variations due to the system motion **and turbulent inflow**".

**Comment 12**  "Line 161: This is a bit confusing. It is possible to understand that the peak-shape parameter is not "systematically varied" like the other parameters, but the text says it "is not varied" - and right after it says that the parameter is varied together with Hs and Tp. Please, reword".
* * *
[1]The rank of a value is its position within a sorted list of data points, ordered from smallest to largest.

**Table A1.** Spearman rank correlation calculation example.

| Sample | My std (kN m) | Rank | $z_{ICR}$ std (m) | Rank | $d_i$ |
|---|---|---|---|---|---|
| 1 | 2.65E+03 | 5 | 120.55 | 4 | 1 |
| 2 | 2.66E+03 | 4 | 119.29 | 5 | -1 |
| 3 | 2.87E+03 | 3 | 126.24 | 2 | 1 |
| 4 | 3.21E+03 | 2 | 122.14 | 3 | -1 |
| 5 | 3.72E+03 | 1 | 149.58 | 1 | 0 |

Figure 1: Caption

**Modification 12**   The sentence was changed to avoid misleading message: "The wave peak-shape parameter is set based on the peak period and significant wave height, as recommended in the IEC 61400-3 Annex B (IEC, 2009)".

**Comment 13**   "Line 165: Missing a reference for the metocean data".

**Modification 13**
"Each design variant is simulated in 10 environmental conditions, listed in Table 2, representing binned metocean data obtained for **the Scottish sectoral marine plan for the NE8 site (Scottish government, 2020)** (these bins account for 98.4 % of cumulative probability)".

**Comment 14**   "Line 171: Is it possible to bring Figure 16 to this section? Then you save the reader from scrolling several pages to see the frames, and at the same time the floaters are shown in the section they are supposed to be described. I also recommend you increase the axes in the figure - they are much smaller than the floaters themselves, which can make it a bit difficult to see them".

**Response 14**   Figure 7 was updated with larger coordinate system axes. As for the placement within the paper, it is placed in the *Coordinate system* section which directly precedes the new *Floating platform and turbine characteristics* section, therefore can now be easily referred to when reading either of the short sections.

**Comment 15**   "Table 2: Can you include the peak-shape factor for groups B and G?".

**Modification 15**   A column was added to Table 5 (tables renumbered as new tables added).

**Comment 16**   "Line 196: Do you assume a constant current profile? Please clarify".

**Response 16**   The sub-surface current follows a power law. We vary the velocity at still water level while leaving the power low exponent unchanged at value 1/7.

**Modification 16**   We modified the text of Section 3.1 Study setup: "Group F adds the effect of the subsurface current speed (the power low exponent is kept at the value 1/7)".

**Comment 17**   "Line 202: It is not very easy to interpret the results, if I am only to look at Fig 2. Are you able to make a plot of the trajectory of the ICR, for one cycle of one of the regular waves, to illustrate a bit better how the ICR moves? You could for example plot also the floater CG trajectory, in the same figure. Very important to show the coordinate-system axes, too".

**Response 17**   The plot of the ICR trajectory during one pitch period in regular waves is provided in Figure 3 and discussed in Section 2.1. The coordinate system is illustrated in Figure 7 as defined

**Table 5.** Impact of varied environmental loading on the ICR – conditions considered. $\gamma$ – peak shape factor.

| | Wind | | | | Waves | | | | Current |
|---|---|---|---|---|---|---|---|---|---|
| Group | Type | $V_s$ | $TI$ | Shear | Type | $H_s$ | $T_p$ | $\gamma$ | $V_c$ |
| (-) | (-) | $(\mathrm{m\,s^{-1}})$ | (-) | (-) | (-) | (m) | (s) | (-) | $(\mathrm{m\,s^{-1}})$ |
| A | - | - | - | - | Reg | 1.87 | 5–17.0 | - | - |
| B | - | - | - | - | JONSWAP | 1.87 | 7.47 | 1.0 | - |
| C | Steady | 3–25 | - | 0.2 | Reg | 1.87 | 7.47 | 1.0 | - |
| D | Steady | 11.4 | - | 0.1–0.4 | Reg | 1.87 | 7.47 | 1.0 | - |
| E | Turbulent | 11.4 | 0.1–0.3 | 0.2 | Reg | 1.87 | 7.47 | 1.0 | - |
| F | - | - | - | - | Reg | 1.87 | 7.47 | 1.0 | 0.5–1.2 |
| G | Turbulent | 11.4 | 0.17 | 0.2 | JONSWAP | 1.87 | 7.47 | 1.0 | 0.85 |

in Section 2.4. The plot of the centre of gravity trajectory is shown in the figure below. We decided not to include this plot in this paper as the position of the centre of gravity only partly impacts the ICR, and does not fully explain its behaviour therefore would not be very informative in the context of the results discussed but could lead to confusion.

[Figure]

**Comment 18**    "Line 204: The longest period considered is 17.0 s. It is hard to believe that these

waves will excite surge (Tn = 125 s) very significantly. Even the heave natural period (31 s) is a bit far from that. What is the pitch natural period? If it is not much longer than 17 s, could it be that the increased surge motion is actually caused by coupling with pitch?".

**Response 18**   Thank you for your careful assessment of the discussion. We did not present the results for periods higher than 17 s as such long waves are rare in real sea states, based on the metocean data. We did analyse the cases of 31 s (pitch natural period) and 125 s (surge natural period), and these resulted in very wide ICR distributions, as presented in the figure included below.

[Figure]

The mechanism leading to the increase of ICR distribution width is the increase of the system response with the exciting frequency decreasing from very low towards the region of the natural frequencies of the system. As seen in the plots below, the surge, heave, and pitch RAOs increase between the periods of 5 s and 17 s, therefore any given wave amplitude will exert higher motions at 5 s than at 17 s, in all rigid body modes.

Following the comment about the wave periods being far below the surge natural period, we performed a more in-depth analysis of the velocities in two points on the structure: z = −120 m and z = −12 m, as considered for ICR calculation, and concluded that two different mechanisms drive the ICR distribution width, one mechanism close to the pitch natural period, and a different one close to the surge natural period, as detailed in the modified text.

**Modification 18**   We updated the discussion in Section 3.2 accordingly: "Two mechanisms are important: i) Difference in the horizontal velocity's increase in two points on the structure due to increase in pitch motion as the wave period increases towards 31 s, and ii) Higher horizontal velocity in both points due to increased surge motion as the wave period approaches 125 s.

More specifically, the ICR is computed based on the intersection of the normals to the velocity vectors at two points on the structure located at $z_A = -120$ m and $z_B = -12$ m. When the wave period approaches the pitch natural period (31 s), the pitch motion increases. Higher pitch motion about a point not coinciding with the midpoint between A and B induces a higher contribution to the horizontal velocity at the point further away from the centre of rotation. With higher pitch motion, the imbalance between the two points becomes more pronounced, and the ICR shifts either higher or lower, depending on the direction of the pitch rotation. This leads to the ICR fluctuating over a wider range.

The second effect becomes more prominent when the wave period is closer to the surge natural

period (125 s). In this case, higher surge motion imposes the same increase of the horizontal velocity in both points. Figure 9 illustrates that, for a given rotational velocity, if a significant horizontal translational velocity is superimposed, the ICR, defined as the point where normals to the velocity vectors intersect, shifts either higher or lower depending on the direction of the translation direction and the instant of the periodic motion (i.e., the phase difference between the translation and rotation). This results in both lower and higher values of the ICR, thereby widening the distributions as observed.

Additional figures supporting this discussion are provided in Figures B1–B2 in Appendix B.".

**Comment 19**  "Line 215: Any possible interpretation for the non-monotonic trend?".

**Response 19**  We have not identified reasons for the non-monotonic trend at this time. The mean values of surge and pitch monotonically increase, and the mean of heave monotonically decreases with the mean wind speed increasing. One interpretation could be based on the change of natural frequencies and phase differences between surge/heave/pitch motions due to the mooring system operating at different mean positions. However, a more detailed investigation would be necessary to understand this unique trend not similar to the rigid body motion trends.

**Comment 20**  "Line 224: It seems this change in mooring loads was not mentioned before".

**Modification 20**  "(...) as observed in the previous tests" was removed from the end of the sentence.

**Comment 21**  "Line 228: You say that the current load on the spar is insignificant, compared with other loads. Still, the variation of zICR varies a lot more for the higher current speed. Why?".

**Response 21**  The distribution of the ICR of the FOWT seems to be dependent on the current speed, however, note that the range on the y-axis is very small, compared to the ones in other relevant plots. The difference in $z_{ICR}$ distribution width between the cases of the lowest and highest current

speed is only $3\,\mathrm{m}$, which is insignificant compared to the ranges in the order of $100\,\mathrm{m}$ observed in other loading cases.

**Modification 21**    A footnote was added to clarify this: "Notice the small range on the y-axis".

**Comment 22**    "Figure 17: Apparently there is a typo describing the marker "Waterline R" (should be "Waterline D" instead?)".

**Response 22**    Thank you for noticing that the label was not clear. The data presented in the figure relates to the waterline radius, consistent with the label. However, Table 9 referred to the waterline diameter, which might have caused confusion. Changes were made to Table 9 to make it consistent with the figures.

**Modification 22**    The label in Table 9 was changed from "Waterline D $(d_{WL})$" to "Waterline R $(r_{WL})$", and the corresponding values halved.

**Comment 23**    "Line 334: Lower nacelle acceleration may also be associated with lower tower base bending moment (reduced inertial loads, especially at wave frequency)".

**Modification 23**    The word "simultaneously" was removed to avoid implying that the nacelle acceleration and tower base bending moment are independent (or can be changed independently). Additionally, the casual relationship was noted in a footnote:
"Namely, by maximising the standard deviation of the horizontal coordinate of ICR (i.e., by shifting it towards the positive $x$ direction), one could reduce the nacelle acceleration and the tower base and blade root bending moments. **FOOTNOTE: *Note that the nacelle acceleration is one of the factors contributing to the tower base bending moment, therefore these responses are not independent.***".

**Comment 24**    "Figures 25 and 27: If I understood it correctly, the bars in the figure show only the mean and std deviation for the modified design - but normalized with the values of the original design. If that is correct, then the caption of the figure may be a bit misleading, since the reader can expect to see the data for both designs side-by-side. Please consider rewording".

**Modification 24**    The updated captions read: "Responses of the modified spar/semisubmersible FOWT design. Values normalised with the original design response values".

**Comment 25**    "Tables 12 and 13: There is something strange with the two last lines in the table: the design variables (e.g. line length and heave plate) should not have a mean and a std deviation".

**Modification 25**    Each table was split into two: one with the mean and standard deviation of responses, and one with design variables.

**Comment 26**    "Line 429: You say that changing the ICR mean z-coordinate improved the responses, but that is true only when we ignore the non-practical change in draft, right? Please be as precise as possible in your conclusions".

**Response 26**    This point is no longer relevant after updating the draft according to the reviewer's comments. Namely, when re-computing the results for the semisubmersible design variables variations updating the potential hydrodynamic loads, as suggested, the platforms were also re-ballasted to keep a constant draft. This was previously only achieved in the design case study, but now we implemented the ballast calculation in the results presented in Section 4 Design variables sensitivity as well.

**Modification 26**    Section 4 text modification: "Note that these design variable perturbations may alter the mean heave (draft) of the floating platform due to changes in displacement volume (buoyancy) and/or the weight of the structure. This effect could significantly impact the response of the FOWT and may be undesirable, as the draft is a critical design parameter established early in the design process. It is often constrained by factors such as port depth and manufacturing or assembly capabilities. Therefore, an additional step was implemented whenever the underwater geometry was modified: the floating platform was re-ballasted to maintain the original draft. This not only affected the total weight of the system but also influenced the weight distribution and center of mass".

**Once again, we would like to thank the reviewer for the thorough review of the article and extremely useful feedback. We believe that this helped to make the paper better, and we greatly appreciate your time spent reviewing it and the useful input you shared.**

---

## Author Response (AR2)

**Response to the Reviewer's comments - review 2**

Investigation into Instantaneous Centre of Rotation for Enhanced Design of Floating Offshore Wind Turbines.

Dear Reviewer,

The authors would like to express their gratitude to the Reviewer for the second review of the paper.

**Comment** "In response to my previous comment nr. 5, the authors said that potential theory loads were computed for the different designs, and that the results in sections 4-6 were updated. However, section 2.3 of the updated manuscript still says that the results assume no change in potential theory loads. Also, the discussion in section 4 does not have any mention to updated potential theory loads. This reviewer still believes that the credibility of the results is significantly impaired by the lack of a discussion on how variations in the design parameters could affect the hydrodynamic loads (and thus the ICR). If this assessment has been performed, as indicated in the response, why they are not discussed in the updated manuscript?".

**Response** Thank you for catching the mismatch between the response and the updated manuscript. The original paragraph about the potential coefficients not being updated had been left in the draft by mistake and has now been removed. The potential coefficients are recomputed for each design variant, as reflected in the updated results. The text now clearly describes the hydrodynamic model:
"The semisubmersible platform's offset columns, heave plates, and main column are modelled using a hybrid approach (potential and Morison). The potential coefficients are obtained with pyHAMS BEM solver (NREL, 2024b), as implemented in RAFT (Hall et al., 2022). The slender pontoons and cross-braces (diameter of 1.6m) are treated with a Morison-only approach, with hydrodynamic coefficients listed in Table 1. Second-order hydrodynamics are not considered."
Thank you for your vigilance.